# 'Primary care sensitive' situations that result in an ambulance attendance: a conversation analytic study of UK emergency '999' call recordings

Matthew James Booker, Ali R G Shaw, Sarah Purdy, Rebecca Barnes

Department of Population Health Sciences, Bristol Medical School, Centre for Academic Primary Care, University of Bristol, Bristol, UK

**Correspondence to**
Dr Matthew James Booker;
Matthew.Booker@Bristol.ac.uk

## ABSTRACT

**Objectives** To explore common features of conversations occurring in a sample of emergency calls that result in an ambulance dispatch for a 'primary care sensitive' situation, and better understand the challenges of triaging this cohort.

**Design** A qualitative study, applying conversation analytic methods to routinely recorded telephone calls made through the '999' system for an emergency ambulance. Cases were identified by a primary care clinician, observing front-line UK ambulance service shifts. A sample of 48 '999' recordings were analysed, corresponding to situations potentially amenable to primary care management.

**Results** The analysis focuses on four recurring ways that speakers use talk in these calls. Progress can be impeded when call-taker's questions appear to require callers to have access to knowledge that is not available to them. Accordingly, callers often provide personal accounts of observed events, which may be troublesome for call-takers to 'code' and triage. Certain question formats—notably 'alternative question' formats—appear particularly problematic. Callers deploy specific lexical, grammatical and prosodic resources to legitimise the contact as 'urgent', and ensure that their perception of risk is conveyed. Difficulties encountered in the triage exchange may be evidence of misalignment between organisational and caller perceptions of the 'purpose' of the questions.

**Conclusions** Previous work has focused on exploring the presentation and triage of life-threatening medical emergencies. Meaningful insights into the challenges of EMS triage can also be gained by exploring calls for 'primary care sensitive' situations. The highly scripted triage process requires precise, 'codeable' responses to questions, which can create challenges when the exact urgency of the problem is unclear to both caller and call-taker. Calling on behalf of someone else may compound this complexity. The aetiology of some common interactional challenges may offer a useful frame for future comparison between calls for 'primary care sensitive' situations and life-threatening emergencies.

## Strengths and limitations of this study

► This is the first study using conversation analysis methods to explore emergency ambulance calls for 'primary care sensitive' situations.
► The study uses a relatively small dataset of 48 calls, and the findings may be limited to situations using the specific triage structure studied.
► Even within this focused data set it is possible to identify recurrent interactional practices that may help understand how 'primary care sensitive' situations reach ambulance dispositions.

annum.[1] The majority of calls are no longer for acutely life-threatening situations[2] but are for problems that may more accurately be termed 'urgent care'. Many of these situations could be managed by a timely contact with a primary care provider, but for an array of complex reasons appear to be entering ambulance service workflows instead.[3]

Previous work by this research group has sought to explore some of the reasons for this trend, and challenge the interpretation that this simply represents 'misuse' of emergency services on the part of patients.[4] There is evidence to suggest a range of socio-economic and contextual factors are associated with ambulance use for 'primary care' problems. These include markers associated with deprivation, isolation and complex conceptualisations of how to cope with unforeseen healthcare problems.[4] There is also evidence to suggest that seeking help on behalf of someone else creates a specific and heightened anxiety.[5] This may lead to an overestimation of the urgency of the situation, and may result in the desire for a more immediate resolution than would otherwise be accepted without this sense of added responsibility.

There is also debate in the policy and academic literature about how 'primary care sensitive' conditions should be defined and

## INTRODUCTION

Calls to the UK ambulance service through the emergency '999' system have been rising over the last decade at a rate of 7% per

identified in this setting, recognising that such conditions are a complex product of the disease, illness and health beliefs, context and healthcare system resources.[6] This study uses the term—informed by the Agency for Healthcare Research and Quality definition[7]—to apply to a broad range of clinical conditions and situations where prompt contact with a primary care clinician may potentially prevent the need for hospitalisation or deployment of emergency clinical resources.

Emotional factors strongly influence decision-making 'in the heat of the moment', when faced with an unexpected need for healthcare advice or treatment.[5 8 9] The emotional aspect is a powerful moderator of help-seeking behaviour, and may even over-ride an individual's own perceptions of how they would hypothetically behave in a given circumstance.[10] The entry point into the urgent and emergency care system for many of these situations is therefore a '999' call to the ambulance service.

In the UK, these calls are handled by a non-clinical call-taker, using a scripted and structured computer-based triage system. In the UK, one of two algorithm-based triage systems are used by ambulance services: The Medical Priority Dispatch System (MPDS)[11] and NHS Pathways.[12] Such triage systems are usually weighted in favour of sensitivity to seriously unwell patients at the expense of specificity, due to the implications of failing to identify and prioritise a time-critical situation. Call-takers are regularly audited to ensure precise compliance with the triage protocols and wording. However, the scripted format can create challenges for call-takers, when callers' responses do not necessarily 'fit' with the question structure, or when callers do not understand the relevance of the information being sought.

Sociolinguistic research has shown that even subtle variations in phrasing, word choices and intonation can have significant impacts on the effectiveness of communication in emergency telephone calls (eg, refs 13–16). Recent research has explored the impact of call-takers' linguistic choices on the efficiency of ambulance dispatch in high-acuity cases, including cardiac arrests,[17] and in the detection of agonal breathing.[18] Earlier work has long recognised the communication challenges posed by the pressure of these 'high-acuity' and 'high-stakes' situations,[19] and on-going research confirms the importance of a more detailed understanding of the dynamics between caller and call-taker to improve dispatch efficiency in these circumstances (eg, refs 17 20). However, little work has been undertaken to explore emergency call interactions in lower-acuity situations, such as for 'primary care sensitive' problems. As these situations are far more common—increasingly forming a substantial portion of UK Ambulance Service workload[3]—there are potentially significant resourcing and efficiency implications for the entire system by understanding if and how these situations may be resolved without an ambulance response. The implications of under-triaging time-critical emergencies (so-termed 'failed emergency calls') are clear, but there are also possibly harmful consequences of diverting scarce resources to situations where emergency interventions are not required.

This study aims to use conversation analysis (CA)—a method that sits at the intersection of sociology and linguistics—to explore the triage dialogue in cases where an ambulance was dispatched to a situation that would likely be successfully managed in a primary care setting. Examples of such situations include the treatment of uncomplicated acute infections, minor musculoskeletal injuries, the management of 'flare-ups' of chronic conditions (including mental health conditions) or indeed any broader situation whereby primary care clinician involvement would likely prevent the need for hospitalisation or an emergency clinical response. By working backwards and undertaking close analyses of a sample of '999' calls associated with such contacts, this work seeks to identify common troubles in interactions that occur in a sample of 'primary care problems' receiving ambulance responses, and understand if (and how) opportunities might exist for alternative responses that may fulfil the caller's needs. The concept of 'trouble' in this study refers to problems that speakers experience in understanding or being understood, and that can manifest as barriers to progressing the dialogue or fulfilling the purpose of the talk.[21] This is distinct from the concept of 'troubles telling' (ie, a speaker explaining what their issue or situation is) that is also seen in the CA literature.[22 23]

Analysing how speakers anticipate, react and respond to evolving 'trouble' in conversation is a key element of understanding what actions people are trying to achieve through their talk,[14] and why outcomes may be different than expected. By undertaking an analysis of the communication strategies evident in these calls, it is hoped that some notable practices of interest can be identified in 'primary care sensitive' situations. This will help frame future direct comparison with non-'primary care sensitive' examples, and enhance the understanding of the mechanics of triaging nonemergency cases in emergency medical systems.

## METHODS

### Participants and setting

This study took place in one UK Ambulance Service in the period of September 2016 to January 2017. The service handles approximately 250 000 emergency calls per annum, serving a population of just under 3 million over an area exceeding 20 000 square kilometres. Adult callers with capacity to consent (either the patient, or a proxy caller who contacted the ambulance service on the patient's behalf) were eligible for inclusion in the study if all three of the following criteria were met:

► The caller had dialled the national emergency '999' number and asked for an ambulance.
► The call had been triaged to receive an emergency ambulance response (of any priority).
► The reason for their call was subsequently deemed to be for a potentially 'primary care sensitive' situation.

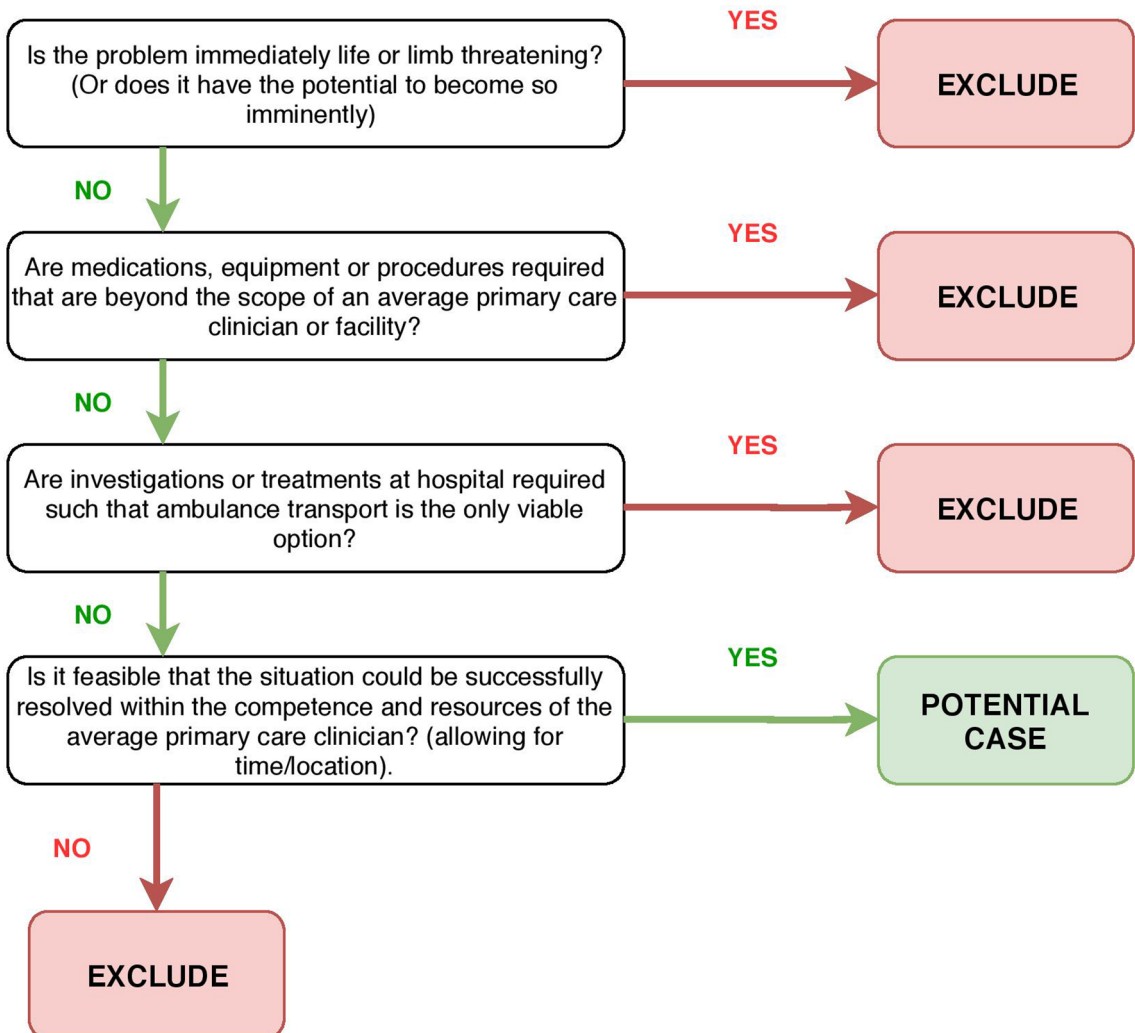

**Figure 1** Indicator criteria for selecting 'primary care sensitive' cases at scene to recruit.

Such situations were identified by the first author—MB, a primary care clinician researcher—who accompanied front-line ambulance crews during routine shifts in a 'non-participant observer' capacity. Predetermined basic indicator criteria (figure 1) and professional judgement were used to identify conditions and situations that would likely be amenable to resolution in a primary care setting. This method of identifying 'primary care' cases was favoured over attempts to use clinical records or routine outcome data, as it was felt that a primary care clinician working at the scene could more accurately assimilate all of the clinical, situational and contextual nuances in real time to make a judgement. The basis for each recruitment was discussed and agreed at study team meetings during the recruitment phase, with recruitment continuing until a broad and diverse representation typical of 'urgent care' presentations had been included, as determined by consensus discussion within the study team and study advisory panel.

At the conclusion of the ambulance service treatment, the patients (and/or their proxy callers, where appropriate) were provided with information regarding the study. Those who requested further details were subsequently formally consented for the use of the '999' telephone call recording associated with the treatment episode.

This study took place in an ambulance service using the Medical Priority Despatch System (MPDS) for triage.[11] Ambulance service call-takers were advised that the study was taking place through internal organisational communication channels, and provided with an opportunity to ask that recordings involving their voice were excluded from the study data set. No such requests were received. Due to the nature of call-takers' work, regular review of call recordings is an established part of their monitoring and development. Call-takers do not personally identify themselves in any way during the calls, and therefore this process was felt adequate.

### Patient involvement

The study team consulted with an Urgent Care Service Users advisory panel, formed to shape the development of this and related studies, at key project milestones. This panel includes patient and carer representatives who have recently used ambulance services. This panel helped shape the focus and design of the study, including the

review of participant-facing study literature and involvement in the dissemination strategy.

## Study data

Call recordings were retrieved by the Ambulance Service in digital form, and provided to the research team in secure, encrypted format in accordance with a customised data security protocol. Any identifiable information such as names, addresses and dates of birth were redacted at the point of receipt by the irreversible insertion of a continuous tone onto the recording, using the audio processing software Audacity (version 2.1.2). This method of redaction was favoured to preserve fidelity of total call length, pause length and timings throughout the recording. All call recordings were professionally transcribed in detail according to established CA conventions and notation.[24] Table 1 provides a transcription key.

## Analytic approach

CA is a well-established qualitative method that focuses on the close analysis of high-quality recordings of naturalistic data (ie, data that is not itself research generated).[25] Data is analysed in a systematic manner, using observations made case-by-case to evidence claims. CA involves the search for patterned communication behaviours that can be identified as a 'practice' of speaking. To be identified as a practice, a particular communication behaviour must be seen to be recurrent and to be routinely treated by speakers in a way that discriminates it from related or similar practices.[26]

Much early CA work was based on analyses of audio-recorded telephone conversations. Often these were mundane calls between friends and family members, although one of the most influential studies was a set of recordings of calls to a suicide prevention centre (by Harvey Sacks in the 1960s).[27] Since then, CA methods have been successfully applied to explore routine interactions in a wide range of healthcare contexts.[28] This is particularly the case in primary care, where CA methods have been used to understand the structural organisation of the acute care medical visit, and how key tasks and goals are accomplished between clinicians and patients.[29–33] CA has also been used to explore computer-aided telephone triage interactions between callers and nurses (eg,[34]), and calls to helplines.[35] The methodology of CA is built around identifying language patterns in observational and retrospective data. It is, therefore, particularly well suited to exploring interactional patterns during a 999 call for a 'primary care problem', and may help understand practices consequential for the call outcome.

Zimmerman (1992) described the overall structural organisation of the emergency call, relating various elements of the structure to the specific purposes they serve.[16] In particular, the 'interrogative series' is a series of sequences of varying length that serve to reach a mutually acceptable description of the 'issue', and advance progressivity of the call towards the provision of help.[16] In this study it was decided to focus on a defined portion

| Table 1 | Transcription notation key |
|---|---|
| **Transcription notation** | **Meaning** |
| (.) | Just discernable pause |
| (0.3), (2.2) | Timed pause (in tenths of a second) |
| ↑ or ↓ | Onset of momentary notable pitch rise or fall |
| Speaker A: [word] Speaker B: [word] | Onset and close of overlapping talk |
| .hh/hh | Hearable intake of breath/ out breath (may be elongated) |
| wo(h)rd | Laughter within word |
| wor- | Sharp cut-off |
| wo:rd | Stretch of preceding sound |
| (word) | Transcriber's best guess at an unclear word or speech particle |
| () | Unclear speech |
| Speaker A: word= Speaker B:=word | No discernable beat of silence between turns (latching) |
| word | Stress on underlined element |
| WORD | Capitals indicate words spoken hearably louder than surrounding speech |
| >word< <word> | Speed change— inward-facing arrows show faster speech, outward show slower |
| °word° | Words that are spoken more quietly than surrounding speech |
| ((words)) | Transcriber's description of some other sound in the recording for example, ((typing)) |
| DIS: | Dispatcher/call-taker |
| CAL: | Caller |

Adapted from G. Jefferson, 'Transcription Notation'.[24]

of the telephone call, that is, from the problem solicitation and interrogative series to how the nature of the call was determined. This would enable identification of specific interactional practices—including the making and responding to requests—and how these might influence an ambulance dispatch outcome. Each call was analysed, therefore, from the call-taker's use of the phrase 'tell me exactly what's happened', until the phrase 'I'm organising help for you now'. In this data set, these two phrases served as consistent parentheses for the 'business' of the initial triage and dispatch determination. Although there are some variations between UK ambulances services in the sequencing and wording of initial questions depending on the version of the triage platform in use, figure 2

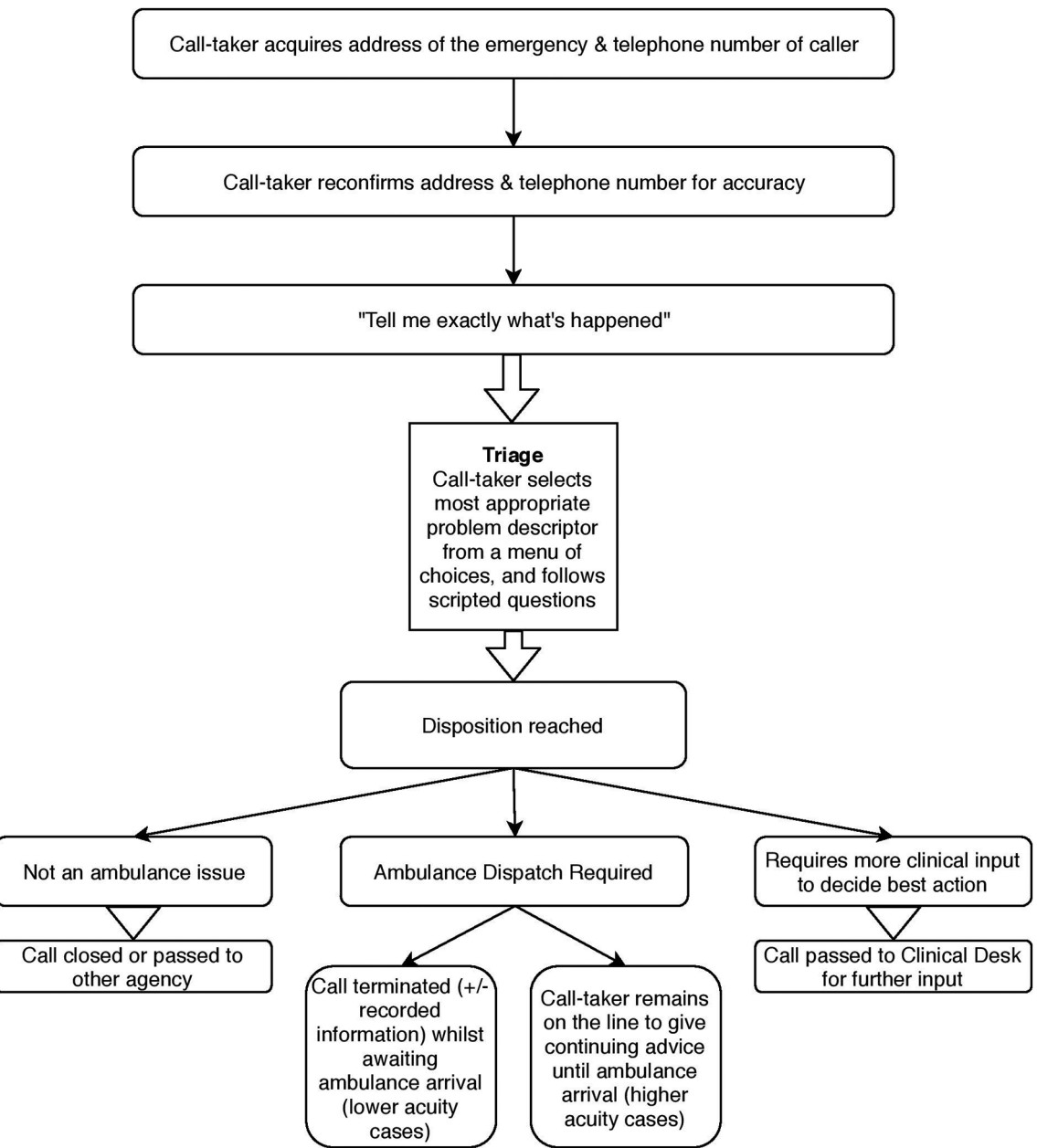

**Figure 2** Overview schematic of the structure of a 999 ambulance call.

provides an outline schematic of the typical call structure in use at the time of this study.

During the analysis, regular data sessions took place involving all of the research team, and a specialist CA methods group within the University of Bristol. This group is diversely comprised of clinicians, social scientists, psychologists and applied CA methodologists. As the aim of this applied research was to focus on interactional elements that may have implications for the understanding the triage of 'primary care sensitive' situations, an initial analysis focused on a subset of 15 recordings. The aim of this preliminary phase was to make detailed observations on a smaller set of cases to identify one or more specific practices of interest, and then build a larger collection by seeking further confirmatory or contradictory examples in the remaining data. The resulting

findings presented below are, therefore, not an exhaustive analysis of *all* practices evident in the data set, but are focused around those of particular relevance to the research focus.

## RESULTS

During the study, 180 hours of front-line shifts were observed by the primary care clinician researcher. In total, 50 eligible cases were recruited in to the study, representing 56.8% of the clinical contacts observed. Of these, 48 '999' call recordings were successfully retrieved and formed the data set for analysis. Due to technical reasons with the way digital recordings are archived, it was not possible to retrieve the remaining two call recordings in a format consistent with the study protocol, and these

**Table 2** Summary of characteristics of recruited cases

| Characteristic | Cases (n=48) |
|---|---|
| Patient mean age (years) | 58.7 |
| Patient age range (years) | 18–92 |
| Patient sex—female | 29 (60%) |
| Has a formal carer | 18 (38%) |
| Not the patient making the 999 call | 31 (65%) |
| Clinical problem | |
| Acute infection | 6 (13%) |
| Breathing problems | 5 (10%) |
| Mental health problems | 5 (10%) |
| Abdominal pain | 4 (8%) |
| Falls, faints and funny turns | 3 (6%) |
| Sickness/gastroenteritis | 3 (6%) |
| Confusion | 3 (6%) |
| Other | 3 (6%) |
| Chronic pain condition flare-up | 3 (6%) |
| Urinary symptoms | 2 (4%) |
| End of life/palliative care problem | 2 (4%) |
| Chest pain | 2 (4%) |
| Musculoskeletal pain | 2 (4%) |
| Skin problems | 2 (4%) |
| Headaches | 2 (4%) |
| Medication problems | 1 (2%) |
| Outcome | |
| Transported to hospital | 14 (29%) |
| Treated at scene—no referrals | 12 (25%) |
| Treated at scene—referred to GP | 17 (35%) |
| Treated at scene—referred to community nursing or social care | 4 (8%) |
| Refused further treatment | 1 (2%) |
| Call made outside of general physician opening hours | 23 (48%) |

cases were therefore excluded from the analysis. Call recordings ranged in length from 2 min 7 s to 21 min 14 s. Table 2 summarises the characteristics of the recruited cases.

The analysis identified four interactional phenomena of particular interest that occurred in these calls. These common areas of difficulty are centred around (1) epistemic positioning, (2) dealing with interactional trouble in the triage, (3) legitimising the contact and (4) interactional resistance. Illustrative cases will be considered below, with reference to the wider CA literature.

### Understanding the questions: the concept of epistemic positioning in the response to 'tell me exactly what's happened'

'Social epistemics' concerns the study of how knowledge is demonstrated, claimed, contested and defended, and is omnirelevant to the analysis of social interaction on the part of participants and analysts alike.[36 37] In the literature on social epistemics, a useful distinction has been made between demonstration or claims of 'type 1 knowledge' (that which is directly experienced, 'first-hand' or observed personally), and 'type 2 knowledge' (that which is known only through indirect means or hearsay).[38 39] Calls for primary care problems were often made on behalf of another and frequently began with a fairly detailed witness account. Extract 1 (box 1) provides an example of how the caller positions their knowledge in response to the opening question, 'tell me exactly what's happened'.

This extract reveals several interesting examples of how the response to the opening question is conveyed in terms of 'type 1 knowledge' of a description of first-hand witnessed phenomena (the details of his symptoms), and the suggestion of a candidate diagnosis at the first point this can be legitimately offered in lines 6–8, that is, *"I think he might have…"*. The caller is in a position of greater access to the patient at this stage, and seeks to convey their first-hand knowledge of the patient in this example with both declarations about the patient's general illness state, and a personal assessment that the situation is bad (lines 5–6, *'I don't like the look of him at all…'*). At this point, the call-taker does not have enough understanding of the situation to proceed with the triage, so a further (unscripted) question is inserted in lines 7–9, '*Okay what so what's happening to him this morning?*' It is interesting to note how the call-taker seeks this further clarification, then immediately repairs the question with *'what symptoms does he have?'*, perhaps recognising that merely using a derivative of the word *'happen'* again will not result in the degree of specific clarity required to progress the triage. It is also relevant to note the shift in tense from the present perfect (ie, what '*has happened*') to the progressive present (ie, what *'is happening'*), indicating a request for current symptoms to be relayed.

This type of self-initiated *self-repa*ir at/after completion of a question can be understood as anticipating potential trouble.[40] In this case the call-taker's self-repair could be a way of responding to early signs of trouble in the exchange by seeking to elicit a clearer description of symptoms that the caller is witnessing *now*.

When callers were not necessarily sure which symptoms, observations or features of concern were most important, they often reverted to expanded witness accounts of the events leading up to the call. Extract 2 (box 2) exemplifies this.

These types of account can be quite difficult to triage, as the principle problem remains unclear for some time.[41] In this example, a clarifying question is needed in line 13. There is a mixing of what the caller has seen ('type 1' knowledge, for example, lines 2, 3, 9, 11–12) and what the caller has indirectly learnt ('type 2' knowledge, eg, 16–17), making it potentially more challenging for the call-taker to reach a confident stance on the knowledge.

## Box 1 Extract 1

```
1   DIS:  thank ↓you::>tellme exactly <what's happe:ned
2         (0.6)
3   CAL:  .hh u:::m, (0.6) ↑my::: (0.4) ↑my (0.4) my: (0.4) eighty
4         eight year old fa:::ther is is bee::n rather unwe:ll
5         (0.4) for quite a while and at this morning I don't like
6         the look of him a[t a:ll °I thi]nk he°
7   DIS:                   [o::kay ↑what]
8   CAL:  might have pneumonia or something like ↑tha::t it's
9   DIS:  so ↑what's ↑h[appe]ning ↑to ↑him this morning=
10  CAL:               [he's]
11  DIS:  =what symptoms does he ha::ve
12         (1.0)
13  CAL:  ↑he:::'s (.) ↑not been eati::ng he's incontine::nt. he's
14        he's u:::m (.) bit deliriou::s (0.4) he's u:::m, (0.6)
15        very wea:k (0.8) he's coughi::ng (.) 'ee's 'ad a:
16        chesty::: well not a chesty cough but a>sort of< (0.4)
17        tickly throaty cou:gh
18         (0.8)
19  DIS:  okay
(Call #16: Relative calling, general decline).
```

Nevertheless, the time-bound nature of the ambulance triage contact, and the need to rapidly progress from the 'unknown' to the 'known' can result in movement towards a triage outcome that—while it might not be precisely correct—it is at least workable. From work on how patients present problems in medical consultations, CA approaches have explored the ideas of 'known' and 'unknown' problems, and how the resulting talk differs in these situations[14]:'Unknown' problems are those where the problem is framed beyond the caller's previous experience, and therefore manifest in a way that the caller doesn't have terminology to succinctly describe. 'Unknown' problems have 'low codeability'.[16] Essentially, it is more difficult for the caller to succinctly convey what

they believe the problem to be, and there may be more of a default to symptoms lists, witness accounts of 'happenings' and 'things they have noticed'. It is easier for a caller to readily report such witness accounts without the need to interpret them beforehand. While it might seem that the opening question *'tell me exactly what's happened'* allows for these low codeability responses, the data set suggests that the triage structure sometimes struggles to manage the diversity of ways that unknown problems are presented. It appears to be particularly a feature of primary care sensitive problems that the exact nature of the problem is often 'unknown'.

Extracts 3 and 4 (box 3) show examples of fairly lengthy opening turns, giving witness accounts of a mixture of

## Box 2 Extract 2

```
1   CAL:  .hh e:::r I got ↑he:re (0.4) and [NAME (1.4)] (sent
2         me her addre::ss) she:: took a while to answer ↑me:
3         actually she (.). hh u::::m>she wouldn't reply on m-<
4         my (0.4) frontdoor ↑ca:ll (0.4) so I [called the]
5   DIS:                                       [didn't ans]wer the
6         door did you say
7   CAL:  no::: [so ↑I] (.) ↑called the o↑ffi:ce (0.4) the offi:ce=
8   DIS:        [okay,]
9   CAL:  =called her by pho:ne the she came down but she ↑looked
10        very un↑stab::le, (.) u:::m she went upstairs to
11        ↑cha:nge>by the< ↑time she came do:wn she still looked
12         (.) again very unsta↑ble,
13  DIS:  what do you mean by unstab:le
14  CAL:  ↑no:t not able to stand u:p er like er she couldn't (.)
15        bala:nce was very bad so she a:sked if she could ↑si:t I
16        asked her if she's not feeling well she said that she's
17        go pain on her stoma:ch sick↑ne::ss,
18  DIS:  [oka:y]
(Call #7:Care staff calling, dizzy spell).
```

**Box 3   Extracts 3, 4 and 5**

```
1   CAL:  er ↑basically my ↑fathe::r, (0.4) u:::m he's been having
2         (.) ↑chest pains all ↑↑ni::ght, (0.4) pretty much nearly
3         all ↑night (0.6). hhh e:rm it's been going like (0.6)
4         like (0.6) like (0.4) sometimes it can be mo:re and
5         sometimes>it can be< ↑le:ss (.)at the moment>he's
6         ↑having< (0.4). hhh the pain a little bit (0.4). hhh
7         (he's got>a little bit<of) ↑ea::se but otherwise i- (.)
8         his a:rms are hurting as ↑↑well
9   DIS:  right (.) okay so are you ↑with ↑him ↑↑no::w?
10  CAL:  yea::h I'm with him no- he's having ↑bu- (0.6) breathing
11        like (0.6). hhhhh just (?) ↑as ↑well,
(Call #24: Relative calling, non-cardiac chest pain)
```

**Extract 4**
```
1   DIS:  thank you ↑tell me exactly what's ↑happe:ned
2   CAL:  e::r I have a daughter that's down syndr↑o:me, hhh (0.6)
3         u:::m she co↑llapsed in the kitchen this morning (0.6)
4         .hh u:m not long ago: a:nd kind of went into:: (.) all
5         her eyes gla::zed and, (0.4) what ↑have you a:::nd,
6         (0.4) she can't put any weight on he::r (.) on her right
7         leg at a:ll. (0.6). hh
[seven lines omitted]
15  CAL:  ↑yeah she's fi::ne ↑she's ↑brea::thi::ng but she's just
16        she's i::n (.) u::m and she's u::m (.) she can't put her
17        foot do:wn (0.6) and it's i::n c- (.) she's in quite a
18        lot of pain with i::t (.). hhh and for her to complain
19        of pai::n (0.8) it must be really ba::d because she's
20        got (.) a tremendous pain thresho:ld
(Call #8: Relative calling, ankle sprain)
```

**Extract 5**
```
1   DIS:  a:nd how far did she ↑↑fa::ll,
2   CAL:  e:::rm (.) I'm not too su:re
3   DIS:  ↑what ↑caused ↑the ↑↑fa:ll?
4   CAL:  I'm not too su:re
5   DIS:  is there any serious ↑blee↑↑di:ng?
6   CAL:  u::m (.) ↑not that I'm aware o:f,
7         (0.6)
8   DIS:  .hh and is she comp↑letely a↑↑wa::ke?
9         (1.0)
10  CAL:  e:::rm yea:h it would appear so,
11  DIS:  >and is she< ↑still on the ↑floo::r?
12  CAL:  e:::rm yes (0.6) I believe so
(Call #3: Carer calling, non-injury fall)
```

symptoms in a manner that suggests the exact nature of the problem is 'unknown' to the caller.

Extract 4 shows how challenging it can be to triage such a contact when the problem is 'unknown' to the caller—does the call-taker assume a faint/collapse (suggested in line 3) and follow the protocol for 'Unconscious/Fainting'? Do they follow the pathway for an injury (lines 6–7), with the protocol for 'Traumatic Injuries'? Or the more general 'Sick Person' as indicated by the pain references (lines 18–20)? The need to select a triage pathway in these cases order to progress the call might be of significance to the eventual outcome of the call to the ambulance service. Extract 5 (box 3) shows a further example of how the 'unknown' is handed by a caller. The caller's

response design indicates a relatively weak epistemic claim ('not that I'm aware of' line 6, 'it would appear so' line 10, 'I believe so' line 12) to each item asked (and the events leading up to the call remain essentially unknown to both parties), but a response is offered in such a format ('not' line 6, 'yeah' line 10, 'yes' line 12) that it is possible to progress the triage relatively promptly.

### Interactional trouble in triage: the 'problems' that scripted triage questions cause

Failing to be able to answer the questions asked is distinct from either a failure to understand them, or a failure to understand the relevance of them.[13] There are examples of all of these in the call recordings within this data set,

**Box 4   Extract 6**

```
1   DIS:  did she ↑faint or nearly ↑fai:nt?
2   CAL:  n- ↑not to my knowl↑edge ↑I: (.) she's only just come to
3         my 'ou:se this morning
4   DIS:  can you ↑check with ↑her if she's ↑fainted ↑or ↑nearly
5         ↑faint↑↑e::d?
6   CAL:  have you ↑fainted or nearly fainted.
7   PAT:  ↑I've been feeling really weak and dizzy:
8   CAL:  she's been really- (0.4) feeling really ↑weak and ↓dizzy:
9   DIS:  oka:y (.)>I just<needher to confirm if she's fainted
10        or nearly fainted
11  CAL:  no: (.) no::
12        (2.4)
13        she's been ↑clo:se (though I think)
14  DIS:  okay so has she ↑nearly ↑fain↑↑te::d?
15  CAL:  well ye:s (0.6) ↑ye::s
16  DIS:  .hh I'm organising help for you now…
(Call #33: Relatively calling, flare of chronic pain condition)
```

and each has different consequences for how the call progresses and the outcome. However, some examples of trouble associated with understanding a question appear to directly influence the triage outcome, as in Extract 6 (box 4). This extract occurs during one of the scripted early questions designed to ascertain the patient's level of consciousness ('did she faint or nearly faint?').

The trouble in responding may have arisen from the use of the alternative question format specifically (line 1), and/or the individual's limited epistemic access to the events. The inadequacy of the caller's response is evident by the pursuit of epistemic primacy at line 4 in the form of a request for action on the part of the caller to seek a confirmed, first-hand account of events. Ambiguity in question design, that is, whether this is a 'polar question' (requiring a 'yes' or 'no' response) or an 'alternative question' (forcing a choice between two candidate events—'*fainted*' or '*nearly fainted*') is the root source of the trouble—the rejection of *both* candidate events would be a dis-preferred outcome in this format.[42]

The call-taker then seeks to resolve the ongoing trouble in line 9 with a request for confirmation, preserving the ambiguous question format. This request seeks information that only the patient has the epistemic right to confirm or disconfirm.[43] This continues to perpetuate the trouble, however, as was apparent some lines previously. The caller doesn't definitively know the answer to this first hand—it is not within his epistemic domain to be able to respond absolutely.

The call-taker can also be seen to indicate tacitly to the caller that this lack of clarity is a direct barrier to progressivity, by the use of the word 'need', suggesting that without this information the triage cannot continue. Interestingly, this does result in a fitted polar response of 'no', in line 11. Perhaps even more interestingly, in line 12——the gap——then results in a retraction of this certainty. The only way the call-taker is able to proceed is by offering the question again but this time in a polar (ie, yes/no) format in line 14, dispensing with the ambiguous format used previously. Notably, this is a 'yes-preferred' polar question,[44] resulting, ultimately, in the ambulance dispatch disposition being reached in the software at that point.

Extract 7 (box 5) also demonstrates trouble arising from an alternative question format —this time an alternative question repair.[42]

In this example, the alternative question that causes the trouble occurs in line 14, and is issued during a multiple repair sequence.[45] This question '*like a fit or like she is cold*' is not actually answered—instead a different question is answered with '*she feels cold*'. The question attempts to clarify a trouble source in the caller's prior narrative description with a choice between alternative understandings—but again the alternative format causes trouble—as the caller states he is unable to differentiate between the two candidate understandings; perhaps because this requires a level of knowledge that he doesn't possess. The repair is repeated in lines 18–19. His disalignment with the question in lines 20 could be considered both a form of resistance, and orientation to a lack of epistemic authority on his part. It is interesting that this display of uncertainty appears to further increase the urgency and the legitimacy of the call.

Alternative questions can be an 'organisational resource' for seeking codeable responses in a triage encounter (particularly when there is a risk that such responses might not be readily forthcoming), in order to progress the prioritisation, and can be designed in the 'to triage' system. They can also be deployed as resources to initiate repair, targeting a trouble source in a caller's narrative description of events. However, in this data set call-takers often needed to deviate to nonscripted polar questions in order to progress away from trouble caused by the alternative question format. The examination of a larger corpus of calls containing scripted alternative questions across a variety of acuities may help to determine if

**Box 5   Extract 7**

```
1   DIS:  >okaythank you sir so< ↑tell me: e↑xactly ↑what's
2         ↑happe::ned
3   CAL:  ↑well, (.)>she was,< (.) she was shaking like a ↑leaf
4         last night and sick, (0.4)>wethought it was<just,
 (didn- kno-) but she's ↑just doing it now she's5        (0.4)
6         been sick a↑gain now (0.4). hhh and e:rm ↑she (.)>she
7         w-< (.)>she was< (.) she was ↑shaking
8   DIS:  okay (.) and whe- (.) when>you sa-<sha- (.) (.) sha-
9         (0.4) sa:y shaking what do you mea:::n
10  CAL:  well she's (.) l- ↑like ↑trem- (0.4) like (.) like like
11        l- li- (.)>lL-<like (0.4) when she's shaking the
12        ↑whole sett↑ees moving like you know ↑what ↑I ↑mean e::r
13        (.) ↑shaking
14 →DIS:  okay (li-) (.) shaking like a ↑fit or shaking like she's
15        co::::ld
16  CAL:  she ↑feels cold an- all
17        (0.4)
18 →DIS:  okay so is it shaking like a ↑fit o:r shaking like she's
19        shiveri::ng.
20  CAL:  like>lL- l-<it's ↑bo- it could be ↑both wa:ys you
21        know she's shivering a:nd, (0.8) she's ↑shaki- if you
22        know what I mean she feels cold and she's shaking ↑I ↑I
23        (0.4). hhh you know not being a medical person I'm ↑not
24        ↑su::re hhh. hh
(Call #2: Relative calling, urine infection)
```

this construction is (a) particularly problematic for third party callers (b) particularly problematic to specific clinical conditions.

### Legitimising the contact as urgent
The concept of legitimacy has been consistently shown to be relevant in studies of healthcare interactions between clinicians and patients.[41] By making an emergency call to the ambulance service, the caller has by their very actions committed themselves to the position that there is a legitimate need for urgent medical attention. This idea has been termed a 'doctorable' problem.[29] Although referring specifically to the primary care setting, the idea transfers very well to seeking ambulance care—the notion that there is a problem '*worthy of medical attention, worthy of evaluation as a potentially significant medical condition, worthy of counselling and… treatment*'.[29]

Callers articulate their legitimacy and genuineness in a variety of ways. The opening question 'tell me exactly what's happened' potentially gives an opportunity space for a caller to deliver an uninterrupted narrative that paints a picture of precisely what they were doing, why they were there, how they came to observe the incident reported, how they reasoned that something problematic might be taking place and what action they took to mitigate this prior to calling.[22 46] While such an account may provide for rich detail and rationale, this may be at the expense of progress to a rapidly 'codeable' determination of clinical urgency.

Extract 2 (box 2) discussed above is an excellent example of this. This example displays all of the elements, including the unprompted explanation of the course of action taken to try and resolve the issue, and the ultimate outcome that an ambulance was recommended by someone with greater authority than the caller (line 27). In this manner, the caller 'hands over responsibility' for the situation to the ambulance service, having clearly been unable to hand it over elsewhere.

Sometimes the proclamation of legitimacy is offered as a description of why alternative avenues have been deemed inappropriate or have not worked. Extracts 8 and 9 (box 6) both describe attempts to access care elsewhere that have failed.

Extreme case formulations are another way that callers can seek to legitimise the contact.[47] This describes the way in which callers seek to set out their symptoms or situation as 'more severe than the average' and therefore in some way more deserving of urgent care. Examples in this data set include the description of the severity of one's own symptoms, as in Extract 10 (box 7). The choice of the phrase '*I cant stand any more*' in lines 2–3 demonstrates how the caller seeks to convey the extreme nature of the circumstance. The offering of '*I'm in pain*' followed immediately by '*I'm in discomfort*' in lines 7–8 may both seek to emphasise the visceral nature of the symptoms, but also suggests that 'pain' and 'discomfort' may mean different things to the caller (perhaps the latter emotional?), each felt worthy or reporting to add to the picture.

There are also a number of examples of how carers and relatives use the extreme case to legitimise the need of the person they care for, as in Extract 11 (box 8).

---

**Box 6    Extracts 8 and 9**

```
1   CAL:  ↑u::::m, (.) ↑right well (0.4) my ↑son i:::s, (0.4) erm
2         ↑hearing voices a:nd he's ↑not ↑very ↑↑good, (0.8) now
3         (.) an ambulance was called yesterday and they took him
4         to [redacted (1.0)] (0.4) a:::::nd ↑th- (.) he was meant
5         to see a psychiatrist this morn↑i::ng,
6   DIS:  ↑yeah
7   CAL:  but the ↑psychiatrist ↑can't ↑see ↑him ↑for ↑a ↑↑week,
8   DIS:  o::kay (.) so are you with ↑him ↑↑no::w?
(Call #41: Relative calling, mental health problems)
Extract 9
1   CAL:  e:::rm (.) yes e::rm my mother who:::'s (0.4) eighty (.)
2         ni::ne (.) u:::::m (.)>is feeling <unwellher ↑heart's
3         raci::ng (.) e::::::::rm (0.4)>feeling< (0.4)>feeling<
4         dizzy: (.) and e:rm (.) sick
5   DIS:  okay
6   CAL:  u:::m (0.4) I've ↑just rung the out of hou:rs (0.4)
7         docto:rs, (0.4) and I spoke to doctor [redacted (2.6)]
8         (.) a::::nd ↑she suggested she said she>wouldn't be
9         able<tocome ou::t
10  DIS:  are you with your ↑mum ↑no:w?
(Call #18: Relative calling, palpitations)
```

The choice of words in line 3 describing the patient in *'a hell of a state'* appears to suggest a genuine and extreme situation. The presentation of a diverse list of symptoms in a fairly condensed fashion also serves to highlight the *'state'* that the patient is in. Even though this situation is clinically 'primary care sensitive', the interaction appears to convey a notable degree of distress.

### Interactional resistance and disalignment

The theories and methods of CA can bring insights into the interactional consequences of mismatched agendas, and provide clues to whether there may have been practical ways to resolve the eventual suboptimal outcomes.

Extract 12 (box 9) focuses on the exchange between a professional carer and the call-taker. The professional carer had just discovered during the morning checks that one of the establishment's residents had died during the night. It became clear after the ambulance attended that—as it was before the GP surgery opening hours—the carer had felt the need to take some form of action to highlight to a relevant official authority that one of the

residents had died, and gain some support in confirming what she knew to be the case.

As the above extract shows, the trajectory the call takes results in cardio-pulmonary resuscitation instructions being given and the caller is talked into delivering resuscitation on what she knows to be a deceased person. The caller demonstrates disalignment with this trajectory, and resists on several occasions. The call-taker acknowledges the resistance in a number of ways, but proceeds with the overall agenda of instructing resuscitation. As has been previously noted, professionals encounter moments where patients resist their actions and institutional agendas.[48] It is clear from this extract that the caller is resisting the institutional agenda of delivering resuscitation, as her own agenda was to gain assistance in confirming the death and discharge her professional obligation to escalate the situation, rather than to expect that the situation be treated as a resuscitation attempt.

Studying examples of interactional resistance reveals a range of explicit and more subtle ways that participants

---

**Box 7    Extract 10**

```
1   CAL:  .hhh er it's my stoma:ch I:::'ve>hada bad<stomach all
2         day (0.4). hhh a:nd um, (0.4). hhh hhh I ↑can't stand any
3         more. hhh hhh (0.6) I I don't know whether I:'m (.)
4         blocked up or what↑ever (>you know<) (.) ↑I've (.)
5         although I opened my bo:wels this ↑↑morning so. hhh
6   DIS:  ↑↑o::↑↑[kay]
7   CAL:  [.hh] i:t's it's it's a:: I'm so:: (0.6). hh I'm
8         in ↑pai:n, I'm in discom↑fort, . hhh hhh
(Call #17: Patient calling, abdominal pain)
```

---

**Box 8 Extract 11**

```
1  CAL:  e:r (.) m- my ↑girlfriend's got th- suffers from
2        endo (.) metrio↑sis (0.6) it's ↑playing up very badly
3        she's having stomach cramps she's in ↑hell of a state
4        (0.6) she's got ↑pai:ns in her le:gs, (1.0)<a:nd (.)
5        she's> (0.6) er> (.) r- ↑bee:n (.) ↑diarrhoea and
6        ↑vomiting with it as well.
(Call #33: Relative calling, chronic pain flare up)
```

can disalign with action agendas in talk.[49] In this example, the caller uses a variety of different devices to resist, including variations of the very explicit *'no'*, *'I can't'*, *'I don't want to'* etc. Nevertheless, the call-taker is able to bring about enough of a degree of alignment for the caller to proceed with resuscitation attempts as instructed. This data is very revealing and suggests that the institutional agenda and implied legitimacy of calling an ambulance are very difficult to deviate from, for both parties, following implementation. Work

around such situations in emergency calls suggests that the caller may disalign with an extended interrogative series, or treat the call-taker's questions as irrelevant, because the institutional reasoning that motivates them is unknown.[14] In the above example, the caller appears to disalign when it becomes clear that the institutional motive (to resuscitate the patient) is different to her own (to discharge her personal responsibility for notifying of a resident's death to someone of authority), and a notably troubled exchange followed.

---

**Box 9 Extract 12**

```
1   CAL:  ↑I:'m ↑just gonna check I think the lady's passed awa:y
2         but I'm not sure.
3   DIS:  ↑o:k[ay]
[10 lines omitted]
14  DIS:  is she a↑wa:::ke?
15  CAL:  ↑no: (.) ↑I I've ↑literally looked at her she feels
16        co:::ld.
17  DIS:  °right° ↑are ↑you ↑able ↑to ↑just ↑go ↑and check for me
18        >to con[firm that<she is] awake and if she is=
19  CAL:  [(?) yeah ]
20  DIS:  =breathi::ng,
21        (18.6)
22  CAL:  no there's no sign of life [at all]
23  DIS:  [no:::] right (.) okay
24  CAL:  .hh hhhh
[15 lines omitted]
40  DIS:  =to do the:n>okay listen<carefu↑lly:, (0.4) I want you
41        to go back into the room and lay her ↑flat on her back
42        on the floo::r and remove any pill↑o::ws,
43  CAL:  but she's on an ↑actual be:d so I ↑really don't wanna be
44        pulling her a↑bou::t,
45  DIS:  okay>but obviously no ↑ifhas passed away<we
46        need to try and help her now best we [ca::n]
47  CAL:  [yeah] right
[12 lines omitted]
60  CAL:  no: I ↑ca:n't to be hone:st she's she's ↑↑not ↑a ↑big
61        lady but she's really hea:↑vy so I ↑↑pu:lled the pillows
62        from under her (0.6). hhh she's definitely a hundred
63        percent gone
64  DIS:  she's ↑o:k[ay]
65  CAL:  [no]:: ↑doubt about i::t hhhh
66  DIS:  right (.) okay we (.)>wif ↑possible not because
67        obviously<u::m, (0.4) we need to try and give her the
68        best possible cha::nce (.) [we need to try and] get her=
69  CAL:  [ye:::s yea:::h ]
Call #2: Professional carer, deceased resident.
```

---

## DISCUSSION

The way institutions perceive, assess and manage risk is at the core of triage in emergency calls. In this context, risk can be classified according to: personal health and safety risks to the caller; risks associated with responding under emergency conditions; risk of causing harm through response (ie, making things worse); risks of technical problems in the triage causing delay; risks associated with access; and risks of error prioritisation (in particularly providing a lower grade response when a higher grade was required).[50]

From a CA perspective, understanding how 'risk' is conveyed, handled and mitigated by both parties in emergency medical calls is crucial. This issue at the juxtaposition of how descriptions are produced so that they appear 'factual', and how these descriptions are then used to perform particular actions.[51] These 'factual' descriptions need to appear solid, neutral and independent of the speaker and merely mirroring some aspect of the world.

This analysis presents a perspective complementary to the literature surrounding triage, risk and decision-making in emergency calls for life-threatening situations. It offers the opportunity to examine the negotiation of risk through talk in calls that were – as defined by the study methodology and case selection criteria described above—ultimately for nonemergency 'primary care sensitive' situations. This contributes to a more nuanced interpretation of risk, which is driven equally by (1) the algorithmic nature of the scripted triage process; (2) the call-taker's use of talk to navigate through blocks in the triage; (3) the caller's narrative description of the situation and (4) how callers use talk to align or disalign with the institutional agenda.

The wider sociological discourse on an individual's ability to assess and manage risk (including the concept of a 'risk society') is highly relevant to this point.[52] [53] There will always be a conflict between the caller's agenda of receiving a resolution *as quickly as possible*, and the institutional agenda of providing a resolution *as quickly as necessary*. Making a call to an emergency service involves the 'regulation and suppression of emotion in the service of a practical task'.[14] Call-takers can find it challenging to engage callers in the collaborative element of the task due to the inherent emotion.[54] This data suggests that the differing perspectives on 'risk', the need for rapidly 'codeable' responses to specific triage questions, and the collective need for progressivity may result in both callers and call-takers using specific interactional practices to advance the call, which may have consequences for how 'primary care sensitive' situations are triaged. In such situations the precise clinical urgency of the problem may be unknown to both parties, yet the need for 'codeable' responses and uncertainty about the other party's attitudes to 'risk' may drive the use of talk practices that ultimately result in ambulance attendance.

Additionally, the impact of 'failed' attempts to acquire resolution elsewhere are omnirelevant to seeking ambulance care, and are evident in a variety of forms in these recordings. These include a mixture of challenges accessing alternative services, re-direction by professionals and responses by other healthcare provides deemed to the caller as unacceptable or incomplete for a variety of reasons. Details of these attempts are often offered by the caller as justification for the contact. Further work is planned to explore how the same situation might be differently described depending on who the answering institution is, and what this might mean for our understanding of 'primary care problems' in the urgent setting.

### Limitations

This analysis has been conducted on a corpus of calls processed through one triage system, MPDS. Findings may not, therefore, be applicable to other triage systems. Further work is planned to compare examples of lower-acuity calls handled through a range of triage systems. Additionally, findings may not apply to non-UK health systems. The determination of the 'primary care sensitive' nature of the contact by a single primary care clinician observer introduces subjectivity to the case recruitment. Attempts were made to provide some objective balance (eg, literature-informed prospective indicator criteria, and discussion of each recruited case by the study team) and as such the authors feel confident that this approach would better identify' true' primary care cases over existing alternative methods based on routine coding or clinical records. However, there does inherently remain some subjectivity.

### CONCLUSIONS

The above analysis presents examples of how the theory and methods of CA can be used to study interactions between '999' ambulance call-takers and callers experiencing 'primary care sensitive' situations. The analysis presented above does not explore all interactional practices evident in the calls, and necessarily presents a small set of examples of the phenomena of interest. Neither are direct comparisons made with non-'primary care sensitive' calls. Despite these limitations, this is the first time—to the authors' knowledge—these methods have been used specifically to investigate talk practices occurring in low-acuity situations that are triaged to dispatch an ambulance. Using CA methods in this way may help provide balance to linguistic approaches exploring high-acuity situations, and pave the way for comparative investigation.

### Implications

This work has implications for triage systems that 'require' highly codeable responses in order to proceed.[34] Certain question formats (ie, alternative questions) appear particularly problematic in some circumstances, yet call-takers manage to adapt their use of talk with 'off-script' subtle variations in order to reach an outcome. Further work should explore whether this question format is generally problematic across different call acuities, and whether such formats should be avoided in triage.

In this data set, the majority of calls were made by someone other than the patient. It is recognised that there are specific challenges with triaging calls made by third parties. This data indicates call triage is also problematic for 'primary care sensitive' situations. Where lines of questioning take callers away from their own epistemic domain, callers may use talk practices to ensure the uncertainly doesn't adversely affect their own goals (ie, by ensuring the recognition of a need for an urgent response is maintained). This may be particularly the case if attempts to access care elsewhere have been met with barriers. Further work is needed to explore the triage structure *specifically* for third-party callers. With increasing evidence that healthcare staff can be successfully trained in more effective talk practices using the lens of CA,[55] there may be ways that call-takers can be trained to design probing questions that are most congruent with third party callers' own epistemic domain, and therefore avoid some of the interactional troubles that can delay triage progression.

Further work is planned to compare examples of triage encounters that do not result in an unnecessary ambulance disposition, and explore whether the use of CA-informed frameworks can be applied to other ambulance service telephone-contacts, such as 'hear and treat' initiatives.

**Acknowledgements** The authors would like to acknowledge Nigel Rees and the Pre-hospital Research Unit at the Welsh Ambulance Service, NHS Trust, for facilitating this study.

**Contributors** MJB is the main author involved in all stages of the design, analysis and write-up. ARGS and SP contributed to the methodological approach, data verification and analysis through regular research meetings. RB provided specialist CA methodological oversight. All authors have approved the final version of the paper ahead of publication.

**Funding** MJB was funded by a National Institute for Health Research Doctoral Research Fellowship. This paper presents independent research funded by the National Institute for Health Research (NIHR). The views expressed are those of the author(s) and not necessarily those of the NIHR, the NHS or the Department of Health.

**Competing interests** None declared.

**Patient consent** Not required.

**Ethics approval** South West (Frenchay) NHS Research Ethics Committee (Reference 15/SW/0307) and appropriate local governance approvals were obtained.

**Provenance and peer review** Not commissioned; externally peer reviewed.

**Data sharing statement** No additional data are available.

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
