## [Reviewer comments · BMJ Open]

ARTICLE DETAILS

TITLE (PROVISIONAL)	'Primary Care Sensitive' situations that result in an ambulance attendance: A conversation analytic study of UK emergency '999' call recordings.
AUTHORS	Booker, Matthew; Shaw, Ali; Purdy, Sarah; Barnes, Rebecca

VERSION 1 – REVIEW

REVIEWER	Dr Stephanie Jones University of central Lancashire, UK
REVIEW RETURNED	10-May-2018

GENERAL COMMENTS	Thank you for the opportunity to review the manuscript entitled 'Why do 'primary care sensitive' situations result in an ambulance attendance? A conversation analytic study of emergency '999' call recordings.' This is an interesting topic but in its current format I have had to recommend that this paper is rejected, as significant revisions are required. Abstract The abstract needs to clearly state the context and background to the study and a clear aim. It isn't clear how 'primary care sensitive' situations are defined. It is also not clear who the sample are and why they have been selected. In what setting did the study take place? Over what time period? How were the data analysed? There are numerous places where the terminology used is very specific to this study and may not be understood on a wider scale. In the results section who are speakers? What is talk? I'm also not sure what is meant by the callers' immediate domain of experience. In what circumstances? What is meant by interactional trouble? What does the institutional project relate to? In the conclusion is it an unknown problem, primary care sensitive problem or both? Again, it isn't clear what interactional problems exist and therefore I can't see the implications of the study findings for practice. Introduction Reference could also be made to calls made by third parties and appropriate literature referred to. A clear definition is needed of 'primary care sensitive' situations. Context is lacking; Within which triage system did this study take place? It would be useful to mention the different triage systems and how these differ/implications of the question structure for your study. Line 34 - an example(s) of situations when cases where an ambulance was dispatched could be managed in primary care. I don't think that 'troubles' accurately reflects the interactions between call taker/caller. Clear aims and objectives are needed. Methods
---

	Participants and setting How many calls does the ambulance service receive? How many calls are for primary care sensitive problems each year? What catchment population does the service cover? What triage system does it use? Over what dates did the study take place? Procedure What was the rationale for the number of calls selected and coded? Was there any cross checking/validation of the selected cases? How many calls would have potentially been included (MB attended 180 hours of front-line shifts) how many cases did this include? Table 1 could be added as supplementary file. The clinical problems listed in table 2 – there is no mention of how they were defined. Analytic approach Did any independent coding take place? Where comparisons made between coders? What was the stance of the coders, I can't see any mention of reflexivity. Results It is mentions, for example, in extracts 8 and 12, that advice had previously been sought by the caller or the call was made out of hours. It would be useful to summarise the data relating to this for all calls and for this information to be added to Table 2. Table 2 there are only 47 clinical problems listed, percentages are needed. It is difficult to read the extracts with the transcription notations included. DIS: CAL: Need to be written in full somewhere. Box 1: Extract.... Are not in boxes? I'm not sure that some of the results have been interpreted fully. For example, on Page 18. Which of the 3 examples given does extract 6 refer to? The caller replies no and then no again later but is asked the same question repeatedly as you discuss on page 19. What about a lack of knowledge to be able to answer the question? Discussion You state that the calls were by definition for lower acuity (page 28) is this defined by MB? Was this agreed/confirmed by any other expert/process? This needs to be expanded upon or mentioned in the limitations. Not all limitations have been acknowledged. For example, how calls were classified as primary care sensitive. It would be useful to mention the different triage systems and acknowledge how this study takes place within one system and the implications this has for the analysis. There are issues other than risk to be discussed – for example, uncertainty about who to contact or a lack of perceived options to due out of hours. Page 29 how do you know that the same although small is typical? The discussion is contradictory in places. The implications for practice need to be more specific.
--	---

REVIEWER	Marine Riou Adjunct Research Fellow, Curtin University, Australia
REVIEW RETURNED	13-May-2018

GENERAL COMMENTS	REVIEW "Why do 'primary care sensitive' situations result in an ambulance attendance? A conversation analytic study of emergency '999' call recordings."
---

Summary

Thank you for inviting me review this interesting qualitative study using the methodology of Conversation Analysis to investigate a corpus of 48 emergency calls for cases which were primary care sensitive but resulted in an ambulance attendance. The topic is a very important one and I am convinced that this type of study is crucial to research on emergency medical dispatch. However, I believe that the discussion goes beyond what the data shows. I think that the way the manuscript is written needs major revisions to appeal to the readership of the journal, and to be clearer about the scope of its claims. Please see detailed comments below.

General comments

Comment #1

The authors seem to claim that a number of interactional features may be characteristic of primary care sensitive calls, as evidenced (for example) by the following sentence in the abstract: "The analysis identified four recurring ways that speakers use talk in calls for 'primary care sensitive' situations" (l.25-26); and by the following sentence in the introduction (l.41-42): "this work seeks to identify common troubles in interactions that may be associated with 'primary care problems'". The interactional features identified by the authors are: (1) callers' limited knowledge, (2) alternative question format, (3) callers expressing urgency, (4) misalignment between caller and call-taker due to competing agendas.

I agree with the authors that investigating these 4 types of interactional issues is a very important endeavor for our understanding of interactional trouble affecting dispatch.

However, I doubt that these 4 features are characteristic of low acuity calls. These features can commonly be found in high priority calls as well, and they may very well be specific of emergency calls in general (when made by a lay caller) – irrespective of the acuity of the situation.[1–3] I see that the authors are planning on conducting a comparative study in the near future. But until this is carried out, I suggest major rewording to clarify the scope of their present claims. I recommend avoiding phrasing implying that the study is looking for defining features of primary care sensitive cases. Besides, I would like to call the authors' attention to previous studies which highlighted the risk of equating certain linguistic styles or emotional states of the caller, to the actual severity of the situation.[4, 5]

Comment #2

The manuscript could be improved if it showed a better understanding of the dispatch process on the call-takers' end – a better knowledge of the constraints and system in which they operate. I understand that the authors made the choice to select cases as assessed when

getting to the patient. However, when analyzing interactional data, it is important to know as much as possible about what bore on that interaction as it unfolded (cf. Enfield's work on enchrony or "conversational time" [6]). Regarding this, I have four concerns as detailed below.

2a) The authors do not indicate which dispatch system is used by the EMS under analysis. Could the authors provide details about the system used (and version)? Based on the excerpts and figures provided, it seems highly likely that what they use is the Medical Priority Dispatch System (MPDS). I assume this is the case for the remaining of my review.

2b) When the authors analyze Excerpt 4, they explain that the call-taker needed "to select a triage pathway", arguing that the call-taker might have hesitated between "faint/collapse", "the pathway for a musculoskeletal injury", and "pain" (l.27-33). Yet, the MDPS does not provide options for the call-taker to choose a chief complaint such as "musculoskeletal injury" or "pain".

2c) Another example of unfamiliarity with the call-takers' side of the calls, is that the authors do not often indicate which call-taker turns are scripted turns that they have to deliver. This can be problematic for analysis, as these scripted turns are not entirely occasioned by the caller's prior turns. Knowing what is scripted as part of the MPDS is crucial to differentiate what is a resource and what is an institutional requirement. Can a scripted question be considered a resource to manage interactional trouble – when it would have been asked no matter what?

2d) The authors assume a retrospective position where they know what condition the patient was found to be in when the paramedics arrived. But this was not known information at the time of the calls. Case in point: I find it quite problematic that a call for chest pains was included in the dataset, and I would assume that the local EMS mandates a priority response to such calls.

Comment #3
Could the authors add arrows in the transcripts to indicate which turns are targeted by their analysis? For example, p.18 l.17, it is unclear whether the authors are referring to l.1 or l.4 in the transcript of Extract 6 when they write: "This extract occurs during one of the standard early questions designed to ascertain the patient's level of consciousness"

Other comments
Comment #4
I am not sure that the title captures the gist of the paper. The study did not focus so much on the reasons why some cases were dispatched an ambulance, but rather, on the interactional features of these calls. Would the authors consider rewording without using the word "why"?

INTRODUCTION

Comment #5

Could the authors provide a reference to support the claim made in the following sentence (l.23) (especially the second part of the sentence): “the emotional aspect is a powerful moderator of help-seeking behaviour, and may override an individual’s own perceptions of how they would hypothetically behave in a given circumstance.”

Comment #6

Could the authors give one or two examples for the type of situation to which they refer l.35-36: “This study, therefore, sought to focus specifically on cases where an ambulance was dispatched to a situation that would likely be successfully managed in a primary care setting”

Comment #7

I can see that the authors define “trouble” as the concept is traditionally used in Conversation Analysis, and their definition is very clear: “The concept of ‘trouble’ in this study refers to problems that speakers experience in understanding or being understood, and that can manifest as barriers to progressing the dialogue or fulfilling the purpose of the talk” (l.47-52). However, the term “trouble” has been used in the two different ways in the literature on emergency calls: “trouble” can refer to “troubles telling” (i.e. caller saying what the situation is), [7–9] or it can refer to interactional trouble (i.e. difficult dialogue between the caller and call-taker) [10, 11] Given that the study will interest readers unfamiliar with the conversation analytic literature, I think it could be useful to mention this ambiguity and clarify it.

METHODS

Comment #8

“The call had been triaged to receive an emergency ambulance response” (l.31): I wonder whether this means that all cases attended by the EMS were candidates for inclusion, or whether the authors made a distinction between different levels of priority response. My reason for asking this is the following. Could one argue that the issue with primary care sensitive cases is when there are sent a high priority ambulance (which then becomes unavailable for a more acute situation), rather than any ambulance response at all?

Comment #9

I disagree with some of the glosses for the symbols used in Table 1 (Transcription Notation

Key):

9a) The up and down arrows are usually used to describe momentary upsteps and downsteps, rather than the onset of a pitch movement (involving several syllables). Could the authors either paraphrase the symbols in a different way, or use different symbols? A useful resource is Szczepiek Reed’s textbook [12] on prosody in conversation.

9b) The authors say that a semicolon (“wo:rd”) correspond to the “stretch of preceding sound

(may be elongated)". I find this unclear. Do the authors mean lengthening? If yes, then is it not elongated by definition?

9c) To define the equal symbol at the end of the turn ("word="), the authors write "No discernable pause between turns". From the notation used, I assume that the authors refer to latching. It would be more accurate to use the word "latching", and to define it without using the word "pause", which can be problematic. I would prefer wording such as "no beat of silence", since turns are not followed by pauses by default.

9d) "emphasis": what type of emphasis do they authors mean? I assume it is stress, rather than volume or lengthening for example, as there are already specific symbols for those in the table?

Comment #10
 There is a missing parenthesis l.31 in the sentence: "Often these were mundane calls between friends and family members, although one of the most influential studies was a set of recordings of calls to a suicide prevention centre by Harvey Sacks in the 1960s."

Comment #11
 Could the authors provide more references when they refer to a "wide range of healthcare context" (l.35) in which the CA methodology had been used? I would suggest (for example) Tanya Stivers' work on pediatrics[13] and John Heritage & Jeffrey Robinson' on GP consults[14-16].

Comment #12
 I am not sure the following sentence (l.43-46) is essential to the logical structure of the section:
 "CA is, therefore, an appropriate methodology to explore interactional patterns during a 999 call for a primary care problem, that might be consequential for the call outcome." The fact that previous studies have used CA for emergency calls does not logically entail that it is an appropriate methodology. If the authors would like to argue that CA is particularly well suited to the study of emergency calls, one argument can be that its methodology is geared towards identifying language patterns in observational and retrospective data.

Comment #13
 Unless I am mistaken, the first time that the term "interrogative series" is used (l.50-54), it has not been defined: "In this study it was decided to focus on a defined portion of the telephone call, i.e. from the problem solicitation and interrogative series to how the nature of the call was determined". I would also appreciate a reference to Zimmerman's paper[17] which introduced the term for emergency calls.

Comment #14
 "Each call was analysed, therefore, from the call-taker's opening phrase 'tell me exactly what's happened', until the closing phrase 'I'm organising help for you now'." (p.10, l.3-7 – and

throughout the manuscript)
 I have an issue with calling the line “tell me exactly what’s happened” an “opening phrase”.
 This line does not open the call, as it is not the first line that call-takers deliver. As evidenced from Figure 2, the call opens with a sequence in which the call-takers confirm the address of the emergency. I refer the authors to the work of Penn et al.[18] and Koole & Verberg[19] on openings in emergency calls.

Comment #15
 The authors write that they analyzed the calls up to when call-takers say “I’m organising help for you now”. To my knowledge, this line is not a mandatory sentence as part of the MDPS script, and so call-takers do not always deliver it. Did the authors encounter calls in which the call-taker did not deliver the line, and if so, how did they determine the cut-off point for their analysis?

Comment #16
 The authors write: “These two phrases served as consistent parentheses for the ‘business’ of the triage process” (p.10, l.7-9). But doesn’t the triage process continue throughout the call, with call-takers reconfiguring triage based on new information which arises, for example during first aid instructions? Maybe a term such as “initial dispatch” would be more accurate.

RESULTS

Comment #17
 The authors write: “In the literature on social epistemics, a useful distinction has been made between demonstration or claims of ‘type 1 knowledge’ (that which is directly experienced, ‘first-hand’ or observed personally), and ‘type 2 knowledge’ (that which is known only through indirect means or hear-say” (p.12 l.22-28)
 I doubt that the medical audience will understand the term “social epistemics” easily. Could the authors connect their definition with the previous sentence more clearly, so that the term is explicitly defined the first time it is introduced?

Comment #18
 The authors write: “This extract reveals several interesting examples of how the response to the opening question is conveyed in terms of ‘type 1 knowledge’ of a description of first-hand witnessed phenomena (the details of his symptoms), and an early offer of a candidate diagnosis in lines 6-8, i.e. “I think he might have”.” (p.13, l.21-26). I do not understand why the authors call this an “early” candidate diagnostics. From the point of view of callers, this is the first “official” slot in the script where they can offer a candidate diagnostic, and many callers offer that at this point in the calls.

Comment #19
 I do not entirely agree with the following sentence: “There is an implicit understanding that the caller is in a position of greater knowledge at this stage” (p.13, l.27-29). While the caller is

the one to have physical access to the patient, the call-taker is the participant that has the expertise. So we are in a situation with a two-way epistemic asymmetry, where knowledge and the rights to that knowledge are negotiated by the participants, which can explain the low epistemic claim of the caller, with the hedge "I think" and the past modal auxiliary "might".

Comment #20

Could the authors add two sets of inverted commas in the following sentence: "It is also relevant to note the shift in tense from what has happened to what is happening, indicating a request for current symptoms to be relayed." (p.13, l.48-58)

Comment #21

The authors write: "In this case the call-taker's self-repair could be a way of responding to early signs of trouble in the exchange by seeking to elicit a clearer description of symptoms that the caller is witnessing now. This potentially makes it 'easier' for the caller to provide the information the call-taker 'needs' to progress the triage" (p.14, l.2-9). I am not sure I understand in what way the progressive present (BE+V-ING) makes it "easier" for the caller.

Do the authors refer to cognitive processes?

Comment #22

I am confused about the structure of the following sentence – what is implied? "The implication of the call-taker's opening question being in the present perfect tense i.e. 'what's happened' is notable." (p.14, l.9-11)

Comment #23

About Extract 2, the authors write: "When callers were not really able to determine the exact nature of the problem, they often reverted to expanded witness accounts of the events leading up to the call." (p.14, l.14-16).

I am not convinced that the expanded account is due to the caller not being able to determine the nature of the problem. In this extract, the caller knows exactly what to describe as a medical problem (pain in the stomach). The fact that they preface it with a long description does not mean that they do not know.

I have a similar issue with another sentence that the authors write later on: "These types account can be quite difficult to triage, as the actual nature of the problem remains unclear for some time." I do not agree that the caller was talking about the problem in an unclear manner. The caller spent some time giving contextual details, but then what they said was clear enough (at that stage) for the call-taker to choose a chief complaint and ask the next questions.

Comment #24

About Extract 3 (p.16): I understand that the chest pains were found to be non-cardiac when the paramedics arrived, but are the authors suggesting that a caller reporting chest pains and

pain in the arm should be identified as primary care if they use terms of uncertainty and do not know what the problem is exactly? Could the call-taker identify with certainty that the case was not an emergency?

Comment #25

About the following sentences (p.17): "The idea of conditional relevance establishes that answers are preferred to non-answers, and that speakers may attempt to make non-answers look like answers in an attempt to progress the talk. In this example the caller doesn't appear to know the answers to the questions, but also appears aware of the need for a response in order to progress the triage."

I feel that some conversation analytic concepts are brought about too abruptly for a medical audience, such as "preference" here. Moreover, I am not sure I see that callers manipulate the format of their answers, as the authors seem to argue. What is it about these caller turns that makes them look like answers? And why do the authors consider that they are actually non-answer responses? Given that the caller needs some time to look and assess the patient before they can answer the call-taker's question, is a delay really a good sign that the answers are non-answers? What is it about the phrase "it appears so" that makes it a non-answer?

Couldn't we argue that it is a confirming answer (to a polar question), but with a low epistemic claim? There seems to be a confusion between response design and strength of epistemic claim.

Could the authors clarify? In the same vein, later in the paragraph, the authors write "In this example the caller doesn't appear to know the answers to the questions" (l.43). I can see in the transcript that the caller expresses uncertainty, but I am not sure I see that the caller expresses absence of knowledge.

Comment #26

What do they authors mean by "standard" question (p.18 l.17) – a scripted question that calltakers always ask, or a turn that they routinely deliver but which is not part of the scripted dispatch protocol?

Comment #27

I find the following sentence structure confusing: "the rejection of both candidates would be a dis-preferred outcome in this format." (p.19 l.11) Does the phrase "both candidates" refer to "polar" and "alternative" question format, or to the candidate events "fainted or nearly fainted"?

Comment #28

"In this dataset, alternative questions can be a resource for seeking codeable responses in a triage encounter" (p.21 l.15): some of these alternative questions, such as "has s/he fainted or nearly fainted" are mandatory scripted questions. Callers are audited on delivered these

turns word for word. So can these scripted questions really be considered a “resource”?

Comment #29

“However, there is evidence in these ‘primary care sensitive’ situations that alternative questions can be particularly problematic, especially for callers who are not themselves the patient.” (p.21 l.23) I am not sure I see how alternative questions would be more problematic in the context of primary care. Can't they occasion the same interactional issues in calls for life-threatening emergencies?

Comment #30

“Approximately two-thirds of calls in this sample showed evidence of trouble around alternative questions” (p.21 l.27-19): without a more in-depth analysis of these questions, I have reservations about this claim. Do the authors refer to a variety of different questions having an alternative format? On what grounds can they claim that these questions created interactional difficulties precisely because of their alternative format? What about the content/meaning of these questions? Maybe they happen to be questions that are hard for lay callers to answer (irrespective of question design), such as prompts to assess the patient’s consciousness or breathing?

Comment #31

About the sentence: “The opening question “tell me exactly what’s happened” gives an opportunity space for a caller to deliver an uninterrupted narrative that paints a picture of precisely what they were doing, why they were there, how they came to observe the incident reported, and how they reasoned that something problematic might be taking place, and what action they took to mitigate this prior to calling.” (p. 22, l.3-13). A narrative containing all these details is precisely what call-takers do NOT want callers to deliver. When some callers do launch into a narrative at this point in the calls, call-takers interrupt them (e.g. saying “but symptoms is s/he having now?”).

Comment #32

Extract 12 p.25: Can a deceased patient really be considered primary care sensitive? And unless proven otherwise, there might be an institutional requirement for the call-taker to treat such a case as cardiac arrest, and thus dispatch a high priority ambulance and provide instructions for resuscitation.

CONCLUSION

Comment #33

P.30 l.39: “Further work is planned to compare examples of triage encounters that do not result in an unnecessary ambulance disposition, and whether it is possible for call-takers to highlight possible ‘primary care sensitive’ cases in real-time during the triage, by identifying disalignment and ‘troubled’ talk in their calls.” As discussed in earlier comments, I find it

unlikely that disalignment and “troubled” talk are distinctive features of low-acuity calls. Making such a claim without a comparison with high acuity calls seems premature.

References

- [1] Garcia AC. ‘Something really weird has happened’: Losing the ‘big picture’ in emergency service calls. *Journal of Pragmatics* 2015;84:102–20.
- [2] Svennevig J. On being heard in emergency calls. The development of hostility in a fatal emergency call. *Journal of Pragmatics* 2012;44:1393–412.
- [3] Tracy SJ. When questioning turns to face threat: An interactional sensitivity in 911 calltaking. *Western Journal of Communication* 2002;66:129–57.
- [4] Eisenberg MS, Carter W, Hallstrom A, Cummins R, Litwin P, Hearne T. Identification of cardiac arrest by emergency dispatchers. *The American Journal of Emergency Medicine* 1986;4:299–301.
- [5] Clawson JJ, Sinclair R. The Emotional Content and Cooperation Score in Emergency Medical Dispatching. *Prehospital Emergency Care* 2001;5:29–35.
- [6] Enfield NJ. Sources of asymmetry in human interaction: enchrony, status, knowledge and agency. In: Stivers T, Mondada L, Steensig J, Eds. *The morality of knowledge in conversation* Cambridge University Press: Cambridge 2011;pp. 285–312.
- [7] Whalen MR, Zimmerman DH. Describing Trouble: Practical Epistemology in Citizen Calls to the Police. *Language in Society* 1990;19:465–92.
- [8] Emmison M, Danby S. Troubles Announcements and Reasons for Calling: Initial Actions in Opening Sequences in Calls to a National Children’s Helpline. *Research on Language and Social Interaction* 2007;40:63–87.
- [9] Weatherall A, Stubbe M. Emotions in action: Telephone-mediated dispute resolution. *Br J Soc Psychol* 2015;54:273–90.
- [10] Tracy K. Interactional Trouble in Emergency Service Requests: A Problem of Frames. *Research on Language and Social Interaction* 1997;30:315–43.
- [11] Murdoch J, Barnes R, Pooler J, Lattimer V, Fletcher E, Campbell JL. The impact of using computer decision-support software in primary care nurse-led telephone triage: Interactional dilemmas and conversational consequences. *Social Science & Medicine* 2015;126:36–47.
- [12] Szczepek Reed B. *Analysing conversation: an introduction to prosody*. Palgrave Macmillan: Basingstoke; New York 2011.
- [13] Stivers T. Non-antibiotic treatment recommendations: delivery formats and implications for parent resistance. *Social Science & Medicine* 2005;60:949–64.
- [14] Heritage J, Robinson JD. “Some” versus “Any” Medical Issues: Encouraging Patients to Reveal Their Unmet Concerns. In: Antaki C, Ed. *Applied Conversation Analysis* Palgrave Macmillan: London 2011;pp. 15–31.
- [15] Robinson JD, Heritage J. Physicians’ opening questions and patients’ satisfaction. *Patient Education and Counseling* 2006;60:279–85.

	[16] Robinson JD, Heritage J. The structure of patients' presenting concerns: the completion relevance of current symptoms. Social Science & Medicine 2005;61:481–93. [17] Zimmerman DH. The interactional organization of calls for emergency assistance. In: Drew P, Heritage J, Eds. Talk at work. Interaction in institutional settings Cambridge University Press: Cambridge 1992;pp. 418–69. [18] Penn C, Koole T, Natrass R. When seconds count: A study of communication variables in the opening segment of emergency calls. Journal of Health Psychology 2016;1–9. [19] Koole T, Verberg N. Aligning caller and call-taker The opening phrase of Dutch emergency calls. Pragmatics and Society 2017;8:129–53.
--	---

VERSION 1 – AUTHOR RESPONSE

Reviewer: 1

Reviewer Name: Dr Stephanie Jones

Institution and Country: University of central Lancashire, UK

Thank you for the opportunity to review the manuscript entitled 'Why do 'primary care sensitive' situations result in an ambulance attendance? A conversation analytic study of emergency '999' call recordings.'

This is an interesting topic but in its current format I have had to recommend that this paper is rejected, as significant revisions are required. Abstract

The abstract needs to clearly state the context and background to the study and a clear aim. It isn't clear how 'primary care sensitive' situations are defined. It is also not clear who the sample are and why they have been selected. In what setting did the study take place? Over what time period? How were the data analysed? There are numerous places where the terminology used is very specific to this study and may not be understood on a wider scale. In the results section who are speakers? What is talk? I'm also not sure what is meant by the callers' immediate domain of experience. In what circumstances? What is meant by interactional trouble? What does the institutional project relate to? In the conclusion is it an unknown problem, primary care sensitive problem or both? Again, it isn't clear what interactional problems exist and therefore I can't see the implications of the study findings for practice.

Thank you for these comments, Dr Jones.

The abstract has been re-drafted. In the paper introduction section, a clear and referenced definition (for the purposes of this paper) of primary care sensitive conditions has been included.

More details on the setting and sampling have been included.

We acknowledged that some of the CA- specific terminology could be problematic for a general readership, and thank you for highlighting some specific examples. (This point has also been considered in relation to Reviewer 2's comments regarding the more gentle introduction of some of the more technical concepts). Some terminology has been rephrased (e.g. institution project, domain of experience) or removed. Some terms (such a 'talk') have been retained, as this is a general CA-related term and is used in other BMJ Open published papers using CA methods (e.g. Syed et al BMJ Open 2017;7:e014260, Murdoch et al BMJ Open 2014;4:e004515).

The paper is predominantly exploring interactional problems arising from 'troubled' talk. In line with these comments and Reviewer 2's comments below on 'troubled' talk, a fuller definition of how this paper defines 'trouble' is included in the main text, with reference to other definitions of 'trouble' used in the CA literature.

Introduction

Reference could also be made to calls made by third parties and appropriate literature referred to.

A clear definition is needed of 'primary care sensitive' situations.

Context is lacking; Within which triage system did this study take place? It would be useful to mention the different triage systems and how these differ/implications of the question structure for your study.

Line 34 - an example(s) of situations when cases where an ambulance was dispatched could be managed in primary care.

I don't think that 'troubles' accurately reflects the interactions between call taker/caller.

Clear aims and objectives are needed.

Thank you for these comments. A clear, referenced definition of primary care sensitive problems has been included, along with some examples. More detail on the context and triage system is included.

The CA-specific term 'troubles' is used in this paper, but in line with Reviewer 2's comments on the importance of differentiating this term from 'troubles telling', a fuller definition has been included, with specific references to how we use the term 'troubles' in its established Conversation Analytic sense. A clearer phrasing of the aims has been included.

Methods

Participants and setting

How many calls does the ambulance service receive? How many calls are for primary care sensitive problems each year? What catchment population does the service cover? What triage system does it use? Over what dates did the study take place?

Details of these elements have been added.

It is not possible to specify what proportion of calls are for primary care sensitive problems, as our previous research has identified that the type of response provided by the ambulance service doesn't necessarily equate to the proportion of primary care sensitive calls (and, indeed, this definition is a significant academic and reporting challenge for which there is currently no consensus). However, for clarity, the proportion of observed cases characterised as primary care sensitive during the study observation period have now been reported in the results section, and we have expanded on detail around the challenges of defining primary care sensitive contacts as per the above comment.

Procedure

What was the rationale for the number of calls selected and coded?

Was there any cross checking/validation of the selected cases? How many calls would have potentially been included (MB attended 180 hours of front-line shifts) how many cases did this include?

Table 1 could be added as supplementary file.

Details of these elements have been included.

We would be happy to delete the table from the paper and move it to a supplementary file if this is editorially preferred.

Analytic approach

Did any independent coding take place? Where comparisons made between coders? What was the stance of the coders, I can't see any mention of reflexivity.

The CA approach is such that data (both transcript and original audio recording) are brought to data sessions attended by multiple CA practitioners and analysed collaboratively. A few more words of explanation, and of the composition of this CA methods group, have been added to hopefully make this clearer.

Results

It mentions, for example, in extracts 8 and 12, that advice had previously been sought by the caller or the call was made out of hours. It would be useful to summarise the data relating to this for all calls and for this information to be added to Table 2. Table 2 there are only 47 clinical problems listed, percentages are needed.

Thank you for this suggestion. We do have data on the timing of all calls, and so a line has been added to Table 2 to summarise the percentage of calls made in the 'out of hours' period. We unfortunately do not have data pertaining to all the calls of whether advice was sought elsewhere or not, so are unable to present this. Thank you also for noting the tallying error – this has been corrected.

Percentages have been included for each of the clinical conditions as suggested.

It is difficult to read the extracts with the transcription notations included.

We do acknowledge this, but we feel very strongly that the transcription notation should be included in the presentation of the extracts as is customary in CA studies.

DIS: CAL: Need to be written in full somewhere.

This is now included in transcription key table.

Box 1: Extract.... Are not in boxes?

The results have been presented this way, as advised, to be typeset accordingly.

I'm not sure that some of the results have been interpreted fully. For example, on Page 18. Which of the 3 examples given does extract 6 refer to? The caller replies no and then no again later but is asked the same question repeatedly as you discuss on page 19. What about a lack of knowledge to be able to answer the question?

There is some over lap with Review 2's comments on knowledge, epistemology and 'known' versus 'unknown' problems: please see response below.

Discussion

You state that the calls were by definition for lower acuity (page 28) is this defined by

MB? Was this agreed/confirmed by any other expert/process? This needs to be expanded upon or mentioned in the limitations.

This has been slightly reworded and more detail given in the methods section about case identification and rationale.

Not all limitations have been acknowledged. For example, how calls were classified as primary care sensitive. It would be useful to mention the different triage systems and acknowledge how this study takes place within one system and the implications this has for the analysis.

Thank you for highlighting this. The 'strengths and limitations' bullet points at the start of the article have been slightly re-worded to reflect this, and a more explicit inclusion of this limitation in the discussion. We have also briefly referred to the various triage systems in the introduction for context.

There are issues other than risk to be discussed – for example, uncertainty about who to contact or a lack of perceived options to due out of hours. Page 29 how do you know that the same although small is typical?

The discussion is contradictory in places.

The implications for practice need to be more specific

We have reworded the discussion and implications sections somewhat, particularly in light of this and reviewer 2's detailed comments below. We do acknowledge that there are many far-reaching issues other than risk that could potentially be discussed. However – and in keeping with Reviewer 2's comments regarding avoiding drawing conclusions that might not have been the focus of the analysis – we have taken the lens of 'risk' as triage is – at its core – essentially about risk determination in primary care sensitive conditions. This is a study of recordings of triage encounters and that has been the analytic lens for the work. We have, however, alluded to some of the other issues that our follow-on work may help understand.

Reviewer: 2

Reviewer Name: Marine Riou

Institution and Country: Adjunct Research Fellow, Curtin University, Australia

REVIEW

“Why do 'primary care sensitive' situations result in an ambulance attendance? A conversation analytic study of emergency '999' call recordings.”

Summary

Thank you for inviting me review this interesting qualitative study using the methodology of Conversation Analysis to investigate a corpus of 48 emergency calls for cases which were primary care sensitive but resulted in an ambulance attendance. The topic is a very important one and I am convinced that this type of study is crucial to research on emergency medical dispatch. However, I believe that the discussion goes beyond what the data shows. I think that the way the manuscript is written needs major revisions to appeal to the readership of the journal, and to be clearer about the scope of its claims. Please see detailed comments below.

General comments

Comment #1

The authors seem to claim that a number of interactional features may be characteristic of primary care sensitive calls, as evidenced (for example) by the following sentence in the abstract: “The analysis identified four recurring ways that speakers use talk in calls for ‘primary care sensitive’ situations” (l.25-26); and by the following sentence in the introduction (l.41- 42): “this work seeks to identify common troubles in interactions that may be associated with ‘primary care problems’”. The interactional features identified by the authors are: (1) callers’ limited knowledge, (2) alternative question format, (3) callers expressing urgency, (4) misalignment between caller and call-taker due to competing agendas. I agree with the authors that investigating these 4 types of interactional issues is a very important endeavor for our understanding of interactional trouble affecting dispatch. However, I doubt that these 4 features are characteristic of low acuity calls. These features can commonly be found in high priority calls as well, and they may very well be specific of emergency calls in general (when made by a lay caller) – irrespective of the acuity of the situation.[1–3] I see that the authors are planning on conducting a comparative study in the near future. But until this is carried out, I suggest major rewording to clarify the scope of their present claims. I recommend avoiding phrasing implying that the study is looking for defining features of primary care sensitive cases. Besides, I would like to call the authors’ attention to previous studies which highlighted the risk of equating certain linguistic styles or emotional states of the caller, to the actual severity of the situation.[4, 5]

Thank you very much for this initial comment, Dr Riou, and your detailed comments and references below. The aims of the work were to explore some interactional issues occurring in calls that end up being for primary care sensitive problems, and present a case that there are meaningful insights that can be gained from a detailed analysis of calls for lower -acuity situations, that may be pertinent for dispatch decisions (and therefore, the system as a whole). The intention was not to claim they these interactional issues are necessarily unique to primary care situations, as indeed we agree that they are likely to be common features of emergency (and other) calls. Although it is our intention to use this work to set the scene for specific comparison work in the follow-on study, this paper has not made any comparison of practices between acuities. However, the intention was to highlight how study of such situations – which we believe is the first attempt to study a set of ‘primary care sensitive’ calls defined in this way and using this methodology – yields meaningful insights. The remarks regarding the phrasing that might suggest these issues are specific to primary care calls is very helpful, thank you. We have reworded a number of sections substantially to make our analytic aims with this dataset clearer, and hopefully addressed this. We also note the references, and in particular are familiar with the Clawson and Sinclair paper and its relevance to a study of this nature, thank you.

Comment #2

The manuscript could be improved if it showed a better understanding of the dispatch process on the call-takers' end – a better knowledge of the constraints and system in which they operate. I understand that the authors made the choice to select cases as assessed when getting to the patient. However, when analyzing interactional data, it is important to know as much as possible about what bore on that interaction as it unfolded (cf. Enfield's work on enchrony or "conversational time" [6]). Regarding this, I have four concerns as detailed below.

2a) The authors do not indicate which dispatch system is used by the EMS under analysis. Could the authors provide details about the system used (and version)? Based on the excerpts and figures provided, it seems highly likely that what they use is the Medical Priority Dispatch System (MPDS). I assume this is the case for the remaining of my review.

This is indeed MPDS. This has been explicitly stated, along with a brief comment around other systems in line with Reviewer 1's comments, above.

2b) When the authors analyze Excerpt 4, they explain that the call-taker needed "to select a triage pathway", arguing that the call-taker might have hesitated between "faint/collapse", "the pathway for a musculoskeletal injury", and "pain" (l.27-33). Yet, the MDPS does not provide options for the call-taker to choose a chief complaint such as "musculoskeletal injury" or "pain".

The intention of this statement was to reflect that the call-taker is required to choose a pathway based on what they believe the caller means by their response, where alternative clinical interpretations may apply. However, we fully accept that the wording does not accurately reflect the MPDS protocol titles available to the call-taker. This has been amended and reworded.

2c) Another example of unfamiliarity with the call-takers' side of the calls, is that the authors do not often indicate which call-taker turns are scripted turns that they have to deliver. This can be problematic for analysis, as these scripted turns are not entirely occasioned by the caller's prior turns. Knowing what is scripted as part of the MPDS is crucial to differentiate what is a resource and what is an institutional requirement. Can a scripted question be considered a resource to manage interactional trouble – when it would have been asked no matter what?

Thank you for this comment. Whilst the analytic approach was not specifically prosodic, the reference to call-taker's use of resources was intended to highlight that there may be prosodic (and other resources) that can be called up even in scripted question delivery. We have reworded some of the text on resources.

2d) The authors assume a retrospective position where they know what condition the patient was found to be in when the paramedics arrived. But this was not known information at the time of the calls. Case in point: I find it quite problematic that a call for chest pains was included in the dataset, and I would assume that the local EMS mandates a priority response to such calls.

Thank you very much for this comment - this is indeed a challenging and complex issue. We agree that the issue of knowing how the final clinical problem relates to what is said and heard in the telephone call is absolutely fundamental to the concept of telephone triage. As such, it was important to us that we included all calls in our corpus that ended up being for a primary care sensitive condition when the EMS arrives, as that is how we have defined our study sample – calls that were for a primary care problem. We agree there is a nuanced (but important) difference between what might sound like the kind of problem that would commonly be dealt with in primary care, and what was actually a clinical diagnosis that could be managed in primary care. To exclude a call for chest pains because it sounds like something that might not be managed in primary care would have undermined the completeness of the corpus, in that ALL calls that ended up being for primary care sensitive situations were included (and indeed, 'chest pain' is a very common presentation to the front desk of a primary care surgery). We do fully understand the issue, however, with suggesting that it might not have been appropriate not to send an ambulance to this

call. We have included in the discussion some reflection around those cases where it may never be possible 'triage out' certain situations – but that runs to the core of the essence of the work.

Comment #3

Could the authors add arrows in the transcripts to indicate which turns are targeted by their analysis? For example, p.18 l.17, it is unclear whether the authors are referring to l.1 or l.4 in the transcript of Extract 6 when they write: "This extract occurs during one of the standard early questions designed to ascertain the patient's level of consciousness"

These have been added.

Other comments

Comment #4

I am not sure that the title captures the gist of the paper. The study did not focus so much on the reasons why some cases were dispatched an ambulance, but rather, on the interactional features of these calls. Would the authors consider rewording without using the word "why"?

Thank you. The title has been revised slightly to reflect this, and in line with the editorial comments regarding setting.

INTRODUCTION

Comment #5

Could the authors provide a reference to support the claim made in the following sentence (l.23) (especially the second part of the sentence): “the emotional aspect is a powerful moderator of help -seeking behaviour, and may override an individual’s own perceptions of how they would hypothetically behave in a given circumstance.”

A reference has been provided of a study that informed this statement, using measurements of anxiety, hypothetical case vignettes and reported own help-seeking behavior.

Comment #6

Could the authors give one or two examples for the type of situation to which they refer l.35- 36: “This study, therefore, sought to focus specifically on cases where an ambulance was dispatched to a situation that would likely be successfully managed in a primary care setting”

A few words of example have been added here, along with a more detailed definition of ‘primary care sensitive’ condition used in this study subsequently.

Comment #7

I can see that the authors define “trouble” as the concept is traditionally used in Conversation Analysis, and their definition is very clear: “The concept of ‘trouble’ in this study refers to problems that speakers experience in understanding or being

understood, and that can manifest as barriers to progressing the dialogue or fulfilling the purpose of the talk” (l.47-52). However, the term “trouble” has been used in the two different ways in the literature on emergency calls: “trouble” can refer to “troubles telling” (i.e. caller saying what the situation is),[7–9] or it can refer to interactional trouble (i.e. difficult dialogue between the caller and call-taker)[10, 11] Given that the study will interest readers unfamiliar with the conversation analytic literature, I think it could be useful to mention this ambiguity and clarify it.

Thank you very much for highlighting this. An expansion of this point has been made, with references as above.

METHODS Comment #8 “The call had been triaged to receive an emergency ambulance response” (l.31): I wonder whether this means that all cases attended by the EMS were candidates for inclusion, or whether the authors made a distinction between different levels of priority response. My reason for asking this is the following. Could one argue that the issue with primary care sensitive cases is when there are sent a high priority ambulance (which then becomes unavailable for a more acute situation), rather than any ambulance response at all?

Thank you for this, this is a very interesting point. In this study, any physical response by the ambulance service – at any priority level – would make the case a candidate for inclusion, on the basis that ambulance care had been delivered rather than primary care in a traditional primary care setting. We do appreciate that there is something potentially very interesting in exploring whether the EMS can (should?) deliver an element of the response to primary care

issues, but that is beyond the focus of this specific element of work. The words 'of any priority' have been added to the inclusion criteria for the avoidance of ambiguity.

Comment #9

I disagree with some of the glosses for the symbols used in Table 1 (Transcription Notation Key): 9a) The up and down arrows are usually used to describe momentary upsteps and downsteps, rather than the onset of a pitch movement (involving several syllables). Could the authors either paraphrase the symbols in a different way, or use different symbols? A useful resource is Szczepek Reed's textbook[12] on prosody in conversation.

Thank you, this detailed comment is noted. For this study we have followed the transcription notation as per the published definitions of Jefferson, cited at the top of the table. We fully note the reference and the other more nuanced ways that this notation can be used, however for this particular piece we feel the above may suggest a more prosodic focus than was actually taken, and therefore have retained the published definition.

9b) The authors say that a semicolon ("wo:rd") correspond to the "stretch of preceding sound (may be elongated)". I find this unclear. Do the authors mean lengthening? If yes, then is it not elongated by definition? 9c) To define the equal symbol at the end of the turn ("word="), the authors write "No discernable pause between turns". From the notation used, I assume that the authors refer to latching. It would be more accurate to use the word "latching", and to define it without using the word "pause", which can be problematic. I would prefer wording such as "no beat of silence", since turns are not followed by pauses by default.

Thank you, we have amended this to make this definition clearer. (The use of 'may be elongated' was intended to refer to the fact that the transcript notation may be

elongated to correspond with the degree of lengthening, e.g. wo:::rd –vs- wo:rd, but to avoid any ambiguity we have rephrased this).

9d) "emphasis": what type of emphasis do they authors mean? I assume it is stress, rather than volume or lengthening for example, as there are already specific symbols for those in the table?

Yes, it is intended to represent stress and this has been amended.

Comment #10

There is a missing parenthesis l.31 in the sentence: "Often these were mundane calls between friends and family members, although one of the most influential studies was a set of recordings of calls to a suicide prevention centre by Harvey Sacks in the 1960s."

Thank you; this has been corrected.

Comment #11

Could the authors provide more references when they refer to a “wide range of healthcare context” (l.35) in which the CA methodology had been used? I would suggest (for example) Tanya Stivers’ work on pediatrics[13] and John Heritage & Jeffrey Robinson’ on GP consults[14–16].

Thank you for suggesting these references; they have been included as further examples of the diverse application of the methodology.

Comment #12

I am not sure the following sentence (l.43-46) is essential to the logical structure of the section: “CA is, therefore, an appropriate methodology to explore interactional patterns during a 999 call for a primary care problem, that might be consequential for the call outcome.” The fact that previous studies have used CA for emergency calls does not logically entail that it is an appropriate methodology. If the authors would like to argue that CA is particularly well suited to the study of emergency calls, one argument can be that its methodology is geared towards identifying language patterns in observational and retrospective data.

The latter point is what was intended by this sentence, which has been reworded to reflect this suggestion, thank you.

Comment #13

Unless I am mistaken, the first time that the term “interrogative series” is used (l.50-54), it has not been defined: “In this study it was decided to focus on a defined portion of the telephone call, i.e. from the problem solicitation and interrogative series to how the nature of the call was determined”. I would also appreciate a reference to Zimmerman’s paper[17] which introduced the term for emergency calls.

An explanation and reference as suggested has been provided.

Comment #14

“Each call was analysed, therefore, from the call-taker’s opening phrase ‘tell me exactly what’s happened’, until the closing phrase ‘I’m organising help for you now’.”

(p.10, l.3-7 – and throughout the manuscript) I have an issue with calling the line “tell me exactly what’s happened” an “opening phrase”. This line does not open the call, as it is not the first line that call-takers deliver. As evidenced from Figure 2, the call opens with a sequence in which the call-takers confirm the address of the emergency. I refer the authors to the work of Penn et al.[18] and Koole & Verberg[19] on openings in emergency calls.

Thank you for the comment, on reflection the choice of the term “opening phrase” does cause some problems in blending ‘lay’ and ‘technical’ terminology and has therefore been removed.

Comment #15

The authors write that they analyzed the calls up to when call-takers say “I’m organising help for you now”. To my knowledge, this line is not a mandatory sentence as part of the MDPS script, and so call-takers do not always deliver it. Did the authors encounter calls in which the call-taker did not deliver the line, and if so, how did they determine the cut-off point for their analysis?

This is the first line in the post dispatch instructions in use at the time of the study, and was present in all of the examples included.

Comment #16

The authors write: “These two phrases served as consistent parentheses for the ‘business’ of the triage process” (p.10, l.7-9). But doesn't the triage process continue throughout the call, with call-takers reconfiguring triage based on new information which arises, for example during first aid instructions? Maybe a term such as "initial dispatch" would be more accurate.

Thank you, this is a valid point and the wording has been amended accordingly.

RESULTS

Comment #17

The authors write: “In the literature on social epistemics, a useful distinction has been made between demonstration or claims of ‘type 1 knowledge’ (that which is directly experienced, ‘first-hand’ or observed personally), and ‘type 2 knowledge’ (that which is known only through indirect means or hear-say)” (p.12 l.22-28)

I doubt that the medical audience will understand the term “social epistemics” easily. Could the authors connect their definition with the previous sentence more clearly, so that the term is explicitly defined the first time it is introduced?

This has been slightly reworded as suggested.

Comment #18

The authors write: "This extract reveals several interesting examples of how the response to the opening question is conveyed in terms of 'type 1 knowledge' of a description of first-hand witnessed phenomena (the details of his symptoms), and an early offer of a candidate diagnosis in lines 6-8, i.e. "I think he might have"." (p.13,

l.21-26). I do not understand why the authors call this an "early" candidate diagnostics. From the point of view of callers, this is the first "official" slot in the script where they can offer a candidate diagnostic, and many callers offer that at this point in the calls.

This term was indeed intended to indicate that it was the first offering of a candidate diagnosis 'at the earliest legitimate point' (thus differentiating it from a symptoms only account). We have rephrased this problematic sentence.

Comment #19

I do not entirely agree with the following sentence: "There is an implicit understanding that the caller is in a position of greater knowledge at this stage" (p.13, l.27-29) . While the caller is the one to have physical access to the patient, the call-taker is the participant that has the expertise. So we are in a situation with a two-way epistemic asymmetry, where knowledge and the rights to that knowledge are negotiated by the participants, which can explain the low epistemic claim of the caller, with the hedge "I think" and the past modal auxiliary "might".

Thank you for this comment, which we agree with. The phrasing has been changed to more clearly articulate that this is an example of two-way epistemic asymmetry.

Comment #20

Could the authors add two sets of inverted commas in the following sentence: "It is also relevant to note the shift in tense from what has happened to what is happening, indicating a request for current symptoms to be relayed." (p.13, l.48-58)

This has been inserted.

Comment #21

The authors write: "In this case the call-taker's self-repair could be a way of responding to early signs of trouble in the exchange by seeking to elicit a clearer description of symptoms that the caller is witnessing now. This potentially makes it 'easier' for the caller to provide the information the call -taker 'needs' to progress the triage" (p.14, l.2-9). I am not sure I understand in what way the progressive present (BE+V-ING) makes it "easier" for the caller. Do the authors refer to cognitive processes?

Thank you for this comment. Whilst this was an indirect reference to cognitive processes, in light of the comment (and the comment below) we feel that expansion on this point would

be potentially confusing and would introduce another complex concept that we have not specifically explored elsewhere in the data analysis. As such we have reworded these two paragraphs.

Comment #22

I am confused about the structure of the following sentence – what is implied? “The implication of the call-taker’s opening question being in the present perfect tense i.e. ‘what’s happened’ is notable.” (p.14, l.9-11)

Thank you, we have re-worded a little as we agree this was a slightly confusing sentence structure.

Comment #23

About Extract 2, the authors write: "When callers were not really able to determine the exact nature of the problem, they often reverted to expanded witness accounts of the events leading up to the call." (p.14, l.14-16) . I am not convinced that the expanded account is due to the caller not being able to determine the nature of the problem. In this extract, the caller knows exactly what to describe as a medical problem (pain in the stomach). The fact that they preface it with a long description does not mean that they do not know.

Thank you for the comment on this. This observation arose from the analysis around accounts blending symptoms with context and witness accounts of preceding events, and how these often resulted in an initial lack of clarity about what the call is about 'right now'. We have reworded this a little.

I have a similar issue with another sentence that the authors write later on: "These types account can be quite difficult to triage, as the actual nature of the problem remains unclear for some time." I do not agree that the caller was talking about the problem in an unclear manner. The caller spent some time giving contextual details, but then what they said was clear enough (at that stage) for the call-taker to choose a chief complaint and ask the next questions.

We have presented this particular example as we feel it does highlight the challenges of working from the most relevant and/or problematic symptom at the time when presented with a blended symptoms and context based initial account. I.e. the "unstable" nature, requiring a clarifying question from the call-taker? The balance problems? The stomach pain and sickness? We have reworded some of the paragraph to hopefully make clearer why this is of interest.

Comment #24

About Extract 3 (p.16): I understand that the chest pains were found to be non-cardiac when the paramedics arrived, but are the authors suggesting that a caller reporting chest pains and pain in the arm should be identified as primary care if they use terms of uncertainty and do not know what the problem is exactly? Could the call-taker identify with certainty that the case was not an emergency?

Thank you, yes we do accept this point and the issues it raises, however feel it is important to present the corpus as analysed. Please see the response to 2d.

Comment #25

About the following sentences (p.17): "The idea of conditional relevance establishes that answers are preferred to non -answers, and that speakers may attempt to make non-answers

look like answers in an attempt to progress the talk. In this example the caller doesn't appear to know the answers to the questions, but also appears aware of the need for a response in order to progress the triage."

I feel that some conversation analytic concepts are brought about too abruptly for a medical audience, such as "preference" here. Moreover, I am not sure I see that callers manipulate the format of their answers, as the authors seem to argue. What is it about these caller turns that makes them look like answers? And why do the authors consider that they are actually non-answer responses? Given that the caller needs some time to look and assess the patient before they can answer the call-taker's question, is a delay really a good sign that the answers are non-answers?

What is it about the phrase "it appears so" that makes it a non-answer? Couldn't we argue that it is a confirming answer (to a polar question), but with a low epistemic claim? There seems to be a confusion between response design and strength of epistemic claim. Could the authors clarify? In the same vein, later in the paragraph, the authors write "In this example the caller doesn't appear to know the answers to the questions" (l.43). I can see in the transcript that the caller expresses uncertainty, but I am not sure I see that the caller expresses absence of knowledge.

Thank you for this detailed comment and these thoughts. We do agree that these are not the best examples of non-answers in the dataset, and instead of introducing the complex concept of preference in an abrupt manner where it might be less relevant, we have removed reference to this and reframed this extract as being an example of knowledge states.

Comment #26

What do they authors mean by "standard" question (p.18 l.17) – a scripted question that call-takers always ask, or a turn that they routinely deliver but which is not part of the scripted dispatch protocol?

This refers to a scripted question. The word scripted is used instead.

Comment #27

I find the following sentence structure confusing: "the rejection of both candidates would be a dis-preferred outcome in this format." (p.19 l.11) Does the phrase "both candidates" refer to "polar" and "alternative" question format, or to the candidate events "fainted or nearly fainted"?

This refers to both candidate events, and has been reworded.

Comment #28

"In this dataset, alternative questions can be a resource for seeking codeable responses in a triage encounter" (p.21 l.15): some of these alternative questions, such as "has s/he fainted or nearly fainted" are mandatory scripted questions. Callers are audited on delivered these turns word for word. So can these scripted questions really be considered a "resource"?

This is intended to refer to where alternative questions are an organisational resource (scripted) or an individual speaker's resource (used spontaneously). This is reworded.

Comment #29

"However, there is evidence in these 'primary care sensitive' situations that alternative questions can be particularly problematic, especially for callers who are not themselves the patient." (p.21 l.23) I am not sure I see how alternative questions would be more problematic in the context of primary care. Can't they occasion the same interactional issues in calls for life-threatening emergencies?

Thank you for these thoughts, which in conjunction with the more general comments at the start of the review have meant that the emphasis of this (and the subsequent) sections have been changed, to better reflect the aim of the paper to highlight practices that are evident (and as such interesting) in primary care sensitive calls,

and therefore worthy of more detailed exploration (Rather than intending to suggest that they are a consequence of (or even necessarily related in any consistent way) the primary care nature.

Comment #30

"Approximately two-thirds of calls in this sample showed evidence of trouble around alternative questions" (p.21 l.27-19): without a more in- depth analysis of these questions, I have reservations about this claim. Do the authors refer to a variety of different questions having an alternative format? On what grounds can they claim that these questions created interactional difficulties precisely because of their alternative format? What about the content/meaning of these questions? Maybe they happen to be questions that are hard for lay callers to answer (irrespective of question design), such as prompts to assess the patient's consciousness or breathing?

Thank you, see above response to 29.

Comment #31

About the sentence: “The opening question “tell me exactly what’s happened” gives an opportunity space for a caller to deliver an uninterrupted narrative that paints a picture of precisely what they were doing, why they were there, how they came to observe the incident reported, and how they reasoned that something problematic might be taking place, and what action they took to mitigate this prior to calling.” (p. 22, l.3-13) . A narrative containing all these details is precisely what call-takers do NOT want callers to deliver. When some callers do launch into a narrative at this point in the calls, call-takers interrupt them (e.g. saying “but symptoms is s/he having now?”).

Thank you for the reflections on this. We were perhaps suggesting that the phrase “tell me exactly what’s happened” might possibly give too much initial freedom to the call-taker, particularly around justification. We have rephrased this section a little to explain why this point was made.

Comment #32

Extract 12 p.25: Can a deceased patient really be considered primary care sensitive? And unless proven otherwise, there might be an institutional requirement for the call-taker to treat such a case as cardiac arrest, and thus dispatch a high priority ambulance and provide instructions for resuscitation.

This is a very interesting point, thank you for raising this, and was the generation of significant discussion within the study team during the analysis. We feel that this could actually be considered a very good example of a possible ‘primary care’ sensitive situation. The accurate determination of ‘obvious death’ or ‘expected death’ (both sub-codes of Protocol 9 – Cardiac Arrest / Respiratory Arrest / Death in the MPDS example) in cases where features of death (as opposed to cardiac arrest) are confidently described by the caller is an opportunity to avoid the futile deployment of multiple EMS resources. The call-taker’s response to cues offered by the caller in such examples is key to arriving at either of two ‘extremes’ in terms of disposition, and are therefore worthy of study alongside other primary care situations.

CONCLUSION Comment #33 P.30 l.39: “Further work is planned to compare examples of triage encounters that do not result in an unnecessary ambulance disposition, and whether it is possible for call-takers to highlight possible ‘primary care sensitive’ cases in real -time during the triage, by identifying disalignment and ‘troubled’ talk in their calls.” As discussed in earlier comments, I find it unlikely that disalignment and “troubled” talk are distinctive features of low-acuity calls. Making such a claim without a comparison with high acuity calls seems premature.

Thank you for this and the earlier comments relating to this. We have amended the wording here to highlight areas that further work will focus on, and how the analysis described here will inform hypothesis generation without – hopefully – over interpreting the findings.

References

- [1] Garcia AC. 'Something really weird has happened': Losing the 'big picture' in emergency service calls. *Journal of Pragmatics* 2015;84:102–20.
- [2] Svennevig J. On being heard in emergency calls. The development of hostility in a fatal emergency call. *Journal of Pragmatics* 2012;44:1393–412.
- [3] Tracy SJ. When questioning turns to face threat: An interactional sensitivity in 911 call-taking. *Western Journal of Communication* 2002;66:129–57.
- [4] Eisenberg MS, Carter W, Hallstrom A, Cummins R, Litwin P, Hearne T. Identification of cardiac arrest by emergency dispatchers. *The American Journal of Emergency Medicine* 1986;4:299–301.
- [5] Clawson JJ, Sinclair R. The Emotional Content and Cooperation Score in Emergency Medical Dispatching. *Prehospital Emergency Care* 2001;5:29–35.
- [6] Enfield NJ. Sources of asymmetry in human interaction: enchrony, status, knowledge and agency. In: Stivers T, Mondada L, Steensig J, Eds. *The morality of knowledge in conversation* Cambridge University Press: Cambridge 2011;pp. 285– 312.
- [7] Whalen MR, Zimmerman DH. Describing Trouble: Practical Epistemology in Citizen Calls to the Police. *Language in Society* 1990;19:465–92.
- [8] Emmison M, Danby S. Troubles Announcements and Reasons for Calling: Initial Actions in Opening Sequences in Calls to a National Children's Helpline. *Research on Language and Social Interaction* 2007;40:63–87.
- [9] Weatherall A, Stubbe M. Emotions in action: Telephone-mediated dispute resolution. *Br J Soc Psychol* 2015;54:273–90.
- [10] Tracy K. Interactional Trouble in Emergency Service Requests: A Problem of Frames. *Research on Language and Social Interaction* 1997;30:315–43.
- [11] Murdoch J, Barnes R, Pooler J, Lattimer V, Fletcher E, Campbell JL. The impact of using computer decision-support software in primary care nurse-led telephone triage: Interactional dilemmas and conversational consequences. *Social Science & Medicine* 2015;126:36–47.
- [12] Szczepek Reed B. *Analysing conversation: an introduction to prosody*. Palgrave Macmillan: Basingstoke; New York 2011.
- [13] Stivers T. Non-antibiotic treatment recommendations: delivery formats and implications for parent resistance. *Social Science & Medicine* 2005;60:949–64.

[14] Heritage J, Robinson JD. "Some" versus "Any" Medical Issues: Encouraging Patients to Reveal Their Unmet Concerns. In: Antaki C, Ed. Applied Conversation Analysis Palgrave Macmillan: London2011;pp. 15–31.

[15] Robinson JD, Heritage J. Physicians' opening questions and patients' satisfaction. Patient Education and Counseling 2006;60:279–85.

[16] Robinson JD, Heritage J. The structure of patients' presenting concerns: the completion relevance of current symptoms. Social Science & Medicine 2005;61:481– 93.

[17] Zimmerman DH. The interactional organization of calls for emergency assistance. In: Drew P, Heritage J, Eds. Talk at work. Interaction in institutional settings Cambridge University Press: Cambridge1992;pp. 418–69.

[18] Penn C, Koole T, Natrass R. When seconds count: A study of communication variables in the opening segment of emergency calls. Journal of Health Psychology 2016;1–9.

[19] Koole T, Verberg N. Aligning caller and call-taker The opening phrase of Dutch emergency calls. Pragmatics and Society 2017;8:129–53.

VERSION 2 – REVIEW

REVIEWER	Marine Riou Adjunct Research Fellow, Curtin University, Australia
REVIEW RETURNED	07-Jul-2018

GENERAL COMMENTS	Summary The revised manuscript answers my previous concerns. I appreciate the efforts made in making changes, and I hear the authors' arguments where they have declined to follow my suggestions. I think that the revised manuscript is much stronger. I only have a few minor comments (see below). Comment #1 P.6 "implications for the entire system by if and how these situations may be resolved". I am not sure about the construction "by of and how" – could the use of the preposition "by" be a typo? Comment #2 P.6 "This study, therefore, aims to use methods informed by sociological and linguistic research to explore the triage dialogue". I think it would add clarity to the non-Conversation Analysis audience to state it clearly right here that what the authors mean by "sociological and linguistic research" is indeed Conversation Analysis. I am not sure it can be expected of all readers of BMJ Open to know that Conversation Analysis as at the intersection of sociology and linguistics.
--

Comment #3

P.6 "This is distinct from the concept of 'troubles telling' (i.e. a speaker explaining what their issue or situation is) that is also seen in the CA literature.[22, 23]"

This seems to be the first occurrence of the term "conversation analysis" in the body of the text, so the abbreviation "CA" needs to be explained here. It seems particularly important to do so in this paragraph, as some readers might think of the abbreviation "CA" for "cardiac arrest", commonly used in the medical literature on emergency calls. However, if the authors accept the suggestion from comment #2, then the abbreviation can be kept here, as it will have been explained at the beginning of the paragraph.

Comment #4

p.9 "The study team consulted at with an Urgent Care Service Users advisory panel"

Between "at" and "with", isn't there a superfluous preposition after the word "consulted"?

Comment #5

p.11 "Zimmerman described the overall structural organisation of the emergency call, relating various elements of the structure to the specific purposes they serve."

Could the authors write the date of the study and provide the reference in this sentence, instead (or in addition to) in the following sentence?

Comment #6

P.12 "Each call was analysed, therefore, from the call-taker's use of the phrase "tell me exactly what's happened", until the closing phrase "I'm organising help for you now"."

I have the same issue with the term "closing phrase" that I had with "opening phrase" in the previous version of the manuscript. It seems problematic to me to call this turn a "closing" turn when it does not close the call. I would suggest simply deleting the word "closing", as the following sentence explains very clearly how the two turns in question work as boundaries.

Comment #7

p.13 Table 2: would it be possible to specify that "mean age", "age range", and "sex" refer to the patient, and not the caller? E.g. writing "patient mean age".

Comment #8

p.16 Could the authors edit the parenthesis from (i.e. what has "happened") and (i.e. what is "happening") to (i.e. what "has happened") and (i.e. what "is happening") so that the entire verb phrases are quoted, and not just the lexical verbs?

Comment #9

p.17 "These types account can be quite difficult to triage"
Typo: "these types of account"?

Comment #10

	p.17 “whilst it might not be precisely correct – it at least workable” Typo: change “it” to “is”. Comment #11 p.17 “the problem is framed beyond the callers previous experience” Typo: change “the callers” to “the caller’s” Comment #12 p.20 “In lines 6, 10 and 12, the caller’s response design indicates a relatively weak epistemic claim to each item asked (and the events leading up to the call remain essentially unknown to both parties), but a response is offered in such a format that it is possible to progress the triage relatively promptly.” To help readers unfamiliar with the concept of epistemic claim, I suggest restructuring the sentence as follows, so that readers can infer what it means from the examples: The caller’s response design indicates a relatively weak epistemic claim (“not that I’m aware of” l.6, “it would appear so” l.10, “I believe so” l.12) to each item asked (and the events leading up to the call remain essentially unknown to both parties), but a response is offered in such a format (“not” l.6, “yeah” l.10, “yes” l.12) that it is possible to progress the triage relatively promptly. Comment #13 p.20 “This extract occurs during one of the scripted early questions designed to ascertain the patient’s level of consciousness.” I suggest that the authors quote the target turn here in parentheses, so that the paragraph can be read independently from the box: “This extract occurs during one of the scripted early questions designed to ascertain the patient’s level of consciousness (“did she faint or nearly faint”).” Comment #14 P.22, Extract 7. The authors have not taken the suggestion to add arrows to indicate which lines are targeted by analysis in the extracts. Can they at least do it for this extract, with an arrow in the margin of l.14 and l.18? Comment #15 p.23 “in this dataset call-takers often need to deviated to non-scripted polar questions” Typo: change to “needed to deviate” Comment #16 p.32 “findings may not apply non-UK health systems” Typo: change to “apply to”?
--	--

VERSION 2 – AUTHOR RESPONSE

Comment 1:

Thank you, we have inserted an omitted word.

Comment 2:

This sentence has been reworded to make the express point that CA sits at the intersection between sociology and linguistics.

Comment 3:

Thank you, the use of the abbreviation at this point is now defined as above in comment 2.

Comment 4:

Superfluous preposition removed.

Comment 5:

The reference and date has been added to this sentence as suggested.

Comment 6:

We understand and accept the comment regarding the use of the word 'closing', and have removed this word as suggested.

Comment 7:

The table has been amended to provide the added clarity that these items are indeed patient age/sex rather than caller.

Comment 8:

The parentheses have been edited to include the entire verb phrase as suggested, with thanks for this suggestion in terms of the added precision this provides.

Comments 9, 10, 11:

These typos are corrected, thank you.

Comment 12:

The sentence has been restructured as suggested to give added clarity to the examples.

Comment 13:

The target turn is now quoted as suggested.

Comment 14:

Arrows have been added to the targeted lines in this extract.

Comments 15 and 16:

Typos corrected, thank you.

VERSION 3 – REVIEW

REVIEWER	Marine Riou Université Lyon 2 Lumière
REVIEW RETURNED	31-Aug-2018
GENERAL COMMENTS	No further comment